# PuzzleFusion: Unleashing the Power of Diffusion Models for Spatial Puzzle Solving

**Sepidehsadat Hosseini, Mohammad Amin Shabani, Saghar Irandoust**[*]**, Yasutaka Furukawa**
Simon Fraser University
{sepidh, mshabani, sirandou, furukawa}@sfu.ca

## Abstract

This paper presents an end-to-end neural architecture based on Diffusion Models for spatial puzzle solving, particularly jigsaw puzzle and room arrangement tasks. In the latter task, for instance, the proposed system takes a set of room layouts as polygonal curves in the top-down view and aligns the room layout pieces by estimating their 2D translations and rotations, akin to solving the jigsaw puzzle of room layouts. A surprising discovery of the paper is that the simple use of a Diffusion Model effectively solves these challenging spatial puzzle tasks as a conditional generation process. To enable learning of an end-to-end neural system, the paper introduces new datasets with ground-truth arrangements: 1) 2D Voronoi jigsaw dataset, a synthetic one where pieces are generated by Voronoi diagram of 2D pointset; and 2) MagicPlan dataset, a real one offered by MagicPlan from its production pipeline, where pieces are room layouts constructed by augmented reality App by real-estate consumers. The qualitative and quantitative evaluations demonstrate that our approach outperforms the competing methods by significant margins in all the tasks. We have provided code and data here.

## 1 Introduction

Spatial puzzle solving demands meticulous reasoning and arrangement of elements within a given space. The classic example of jigsaw puzzles, which many of us have enjoyed as a recreational activity, showcases the challenge and satisfaction of such tasks. Applications of jigsaw puzzles extend beyond mere entertainment, encompassing areas such as the restoration of shattered 2D artwork and documents Das et al. (2017); McBride and Kimia (2003), image stitching in computer vision Hammoudeh and Pollett (2017), and even DNA sequence assembly in genomics Pop (2009); Marande and Burger (2007). In real estate, room layout arrangement has emerged as a compelling spatial puzzle task, where consumers leverage mobile devices to scan individual rooms that are to be arranged into a floorplan.

Spatial puzzle solving poses a considerable challenge even for humans, making it an engaging mental exercise. Addressing this challenge for computational approaches is even more demanding, despite the emergence of deep learning. Current state-of-the-art techniques typically enumerate pairs of aligned pieces, evaluate their compatibility by learning, and employ optimization or search methods to identify the most likely global arrangement Harel and Ben-Shahar (2021); Hoff and Olver (2014a); Shih and Lu (2018). However, these approaches struggle to scale as the complexity of global arrangement increases exponentially with the number of pieces.

This paper makes a breakthrough in spatial puzzle solving with an end-to-end neural architecture based on Diffusion Models Ho et al. (2020), named PuzzleFusion. The surprising discovery of this paper is that a Diffusion Model, typically regarded as a powerful generative model, is an effective

---

[*]Work done while being student at SFU

37th Conference on Neural Information Processing Systems (NeurIPS 2023).

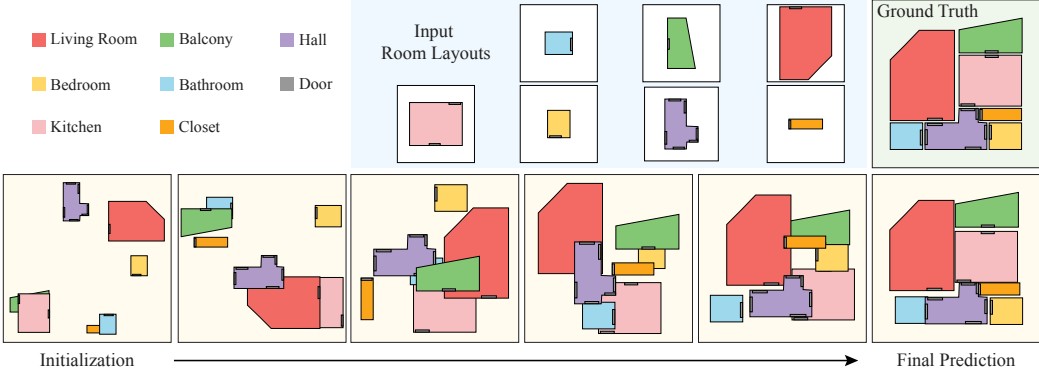

Figure 1: Room layout arrangement is the task of taking a set of room layouts and their corresponding room types as the input and predicting the position and the orientation of each room. The biggest discovery and surprise of this paper is that conditional generation by a Diffusion Model solves this challenging problem.

spatial puzzle solver. PuzzleFusion formulates the spatial puzzle as a conditional generation process where the piece information is the condition.

Concretely, each puzzle piece is a polygonal shape, represented by a sequence of 2D corner coordinates. A piece is associated with task-dependent properties, such as a texture image for pictorial jigsaw puzzles or the room types and door locations for Room Layout Arrangement. The output is a 2D position and an orientation for each piece. The forward diffusion process adds noise to a feature vector initialized with the ground truth position and orientation. The reverse denoising process learns to infer position and orientation subject to the piece shape and property information as a condition.

Qualitative and quantitative evaluations over the three tasks (Cross-cut jigsaw, Voronoi jigsaw, and room layout arrangement) demonstrate that PuzzleFusion outperforms the current state-of-the-art methods by significant margins. The paper also makes dataset contributions by introducing large-scale datasets with ground-truth labels for Voronoi jigsaw and room layout arrangement tasks. Specifically, the room layout arrangement dataset comes from a production pipeline by MagicPlan (https://www.magicplan.app/) consisting of room-layouts and floorplans for 98,780 houses, which we obtained permission to share with the research community.

In summary, this paper makes the following three key contributions: 1) An end-to-end neural architecture based on Diffusion Models for spatial puzzle solving; 2) State-of-the-art performance across three spatial puzzle tasks in terms of accuracy and speed; and 3) The new spatial puzzle datasets including room-layouts and floorplans for 98,780 houses from a production pipeline. We will make all our code and data public.

## 2  Related Work

Spatial puzzle solving is closely related to Structure from Motion (SfM), pose estimation, arrangement learning, and more. The section discusses the related techniques.

**Feature matching** has been successful for the SfM problem Snavely et al. (2006); Li et al. (2020); Lin et al. (2016). The rise of deep neural networks enables more robust feature matching by learning Yi et al. (2018); Sun et al. (2021); Sarlin et al. (2020). However, these techniques require visual overlaps. Our task has little to no visual overlaps between adjacent images.

**Geometry inference** estimates a relative pose between images or partial scans with minimal overlaps by registering inferred or hallucinated geometry. A popular approach learns the priors of room shapes and alternates pairwise alignment and scene completion Lin et al. (2019); Yang et al. (2019). Yang *et al.* Yang et al. (2020) combines global relative pose estimation and local pose refinement

with panoramas. These techniques learn priors of a single room, while our approach learns the arrangements of multiple rooms on a house scale.

**Arrangement learning** is the current state-of-the-art for indoor room layout arrangement. An early work uses windows to align indoor and outdoor reconstructions Cohen et al. (2016). Shabani *et al*. Shabani et al. (2021) use doors to enumerate room arrangements and learn to score each candidate. Their approach is exponential in the number of rooms with many heuristics. Lambert *et al*. Lambert et al. (2022a) uses doors, windows, and openings to create room alignment hypotheses. They utilize depth maps to create top-down views and learn to verify the correctness, improving a run-time from exponential to polynomial. Our end-to-end approach does not enumerate arrangement candidates and makes significant performance improvements. Lastly, an annotated site map and SfM reconstructions are aligned to solve a challenging structure from motion problem Martin-Brualla et al. (2014); Hosseini and Furukawa (2022), which they coined as a "3D jigsaw puzzle".

**Puzzle Solving** has been an engaging research area for a long time Freeman and Garder (1964); Radack and Badler (1982); Markaki and Panagiotakis (2022), ranging from pictorial puzzles with image information Shih and Lu (2018); Le and Li (2019); Toler-Franklin et al. (2010) to apictorial puzzles with only geometry information Goldberg et al. (2002); Harel and Ben-Shahar (2020); Hoff and Olver (2014b, 2013); Harel and Ben-Shahar (2021). In previous studies, heuristic-based methods, such as edge and color matching Wolfson et al. (1988); Nielsen et al. (2008); Chung et al. (1998), have been predominantly used for solving both methods. Recently, deep learning-based methods seek to learn high-level pictorial features Noroozi and Favaro (2016); Li et al. (2021a). However, they suffer from poor geometric reasoning and are limited to simple square puzzles. Consequently, researchers combine learning-based models with heuristics to handle more complex puzzles Le and Li (2019). Nonetheless, these methods still require explicit pairwise comparison of pieces and lack data-driven high-level reasoning . Our method overcomes these limitations through an innovative end-to-end use of Diffusion Models which are usually regarded as powerful generative models.

**Diffusion models** (DMs) are emerging generative models, which slowly corrupt a sample by adding noise Ho et al. (2020); Dhariwal and Nichol (2021); Nichol and Dhariwal (2021a); Shabani et al. (2022), learn to invert the process, and generate a diverse set of samples from noise signals. DMs have established SOTA performances in numerous tasks such as image colorization/inpainitng Song et al. (2020); Nichol et al. (2021), image to image translation Sasaki et al. (2021); Zhao et al. (2022), text to image Ramesh et al. (2022), super-resolution Rombach et al. (2021); Saharia et al. (2021); Li et al. (2021b), image and semantic editing Avrahami et al. (2022); Meng et al. (2022), and denoising Kawar et al. (2022). Recent works use DMs as representation learners for discriminative tasks such as image segmentation Baranchuk et al. (2021); Wolleb et al. (2021). Diffusion inspired models have been used for human pose estimation Shan et al. (2023); Gong et al. (2023), and object placement Wei et al. (2023), however, both tasks do not require sophisticated shape comparisons. While in our case a more advanced and intricate approach is necessary to capture and exploit all details of the shapes.

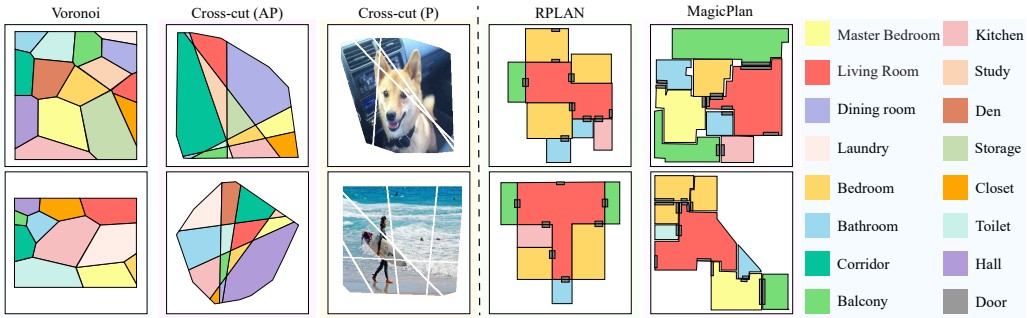

Figure 2: Spatial puzzle datasets for Voronoi and Cross-cut puzzle solving (Left) and room layout arrangement (Right). We consider both pictorial and apictorial versions of the Cross-cut jigsaw puzzle. Note that colors in puzzle solving problems are random and just indicate different pieces.

# 3 Spatial Puzzle Solving Tasks

Spatial puzzle solving involves arranging puzzle pieces, each with its geometry and optional features such as images or categories. In real scenarios, puzzle pieces are affected by erosion, duplication, or loss. This paper investigates three distinct puzzle-solving tasks.

• Cross-cut Jigsaw Puzzle (CJP) generates pieces by making random straight cuts through a larger polygonal shape Harel and Ben-Shahar (2021). Puzzle pieces are convex polygons, each with an arbitrary number of neighboring pieces. The sum of the two adjacent angles at any corner equals $180°$. The application of the technique could be the restoration of shattered artwork.

• Voronoi Jigsaw Puzzle (VJP) generates pieces by randomly sampling points within a predetermined bounding box and extracting the cells of the Voronoi diagram as the pieces. With the lack of the "$180°$ constraint", VJP is significantly more challenging, where no effective solution has been presented in the literature to our knowledge. Voronoi diagrams play a role in biological systems like cell arrangements Bock et al. (2010), potentially useful in studying natural phenomena.

• Room Layout Arrangement (RLA) determines the room arrangement and the corresponding floorplan, offering a key application in real estate. Contrary to the prior tasks, a corner may align along the edge of another piece, expanding the solution space to be explored. Pieces come from the room layout estimation algorithms Shabani et al. (2021); Lambert et al. (2022a) or interactive augmented reality apps used by consumers.

**Task input/output**: The input is a set of $N$ polygons (puzzle pieces in CJP/VJP and room layouts in RLA), each of which is a sequence of corner coordinates forming a 1D polygonal loop. In RLA, a door piece is also given as a line segment with two corners, and a room piece is associated with a room type as a 20D one-hot vector. For a pictorial version of CJP, a piece is associated with a 128D image feature vector obtained from a pretrained auto-encoder. For simplicity, we mix room-corners and door-corners, and use $C_i^r$ to denote the 2D coordinate of the $i$th corner in the $r$th polygon. $\mathcal{T}^r$ denotes the image feature or the room type vector. Please refer to supplementary for details. The output is the position of the piece/room center and the rotation around it (an angle between 0 and $2\pi$ for CJP/VJP and a 4-fold Manhattan rotation for RLA). The center is the average of the corners.

**Metrics**: For CJP and VJP, we adopt metrics used in previous work Harel and Ben-Shahar (2020), namely Overlap, Precision, and Recall. The overlap score is the average IoU of pieces with the ground truth. Precision and Recall are on the connectivity of neighboring pieces. For RLA, we consider two metrics. The first metric is the Mean Positional Error in pixels (MPE) over the rooms Shabani et al. (2021). [2] The second metric evaluates the correctness of the room connectivity in the reconstruction. We borrow a Graph Edit Distance (GED) by Nauata *et al*. Nauata et al. (2021), which counts the number of user edits necessary to fix the connectivity graph. We declare that two rooms are connected if the door centers are within 5 pixels from the two rooms. Note that we have designed task-specific metrics, respecting the methodologies of existing literature while providing a thorough evaluation of our performance.

# 4 Spatial Puzzle Solving as Conditional Generation

Our idea is simple, using a Diffusion Model to "conditionally generate" the correct arrangement, where the input is the center and rotation of room layouts/puzzle pieces and their types and shapes are the conditions. This section explains the forward and the reverse processes.

## 4.1 Forward process

The forward process adds a Gaussian noise to an arrangement. A compact representation would be per-polygon positions and rotations. Instead, we will use a redundant representation, where a center position and a rotation are stored at each corner.

---

[2]Shabani et al. Shabani et al. (2021) used a permissive metric (the availability of the "correct" solution in the k results with a certain error-tolerance) as the task was challenging. We make great improvements and use a standard metric.

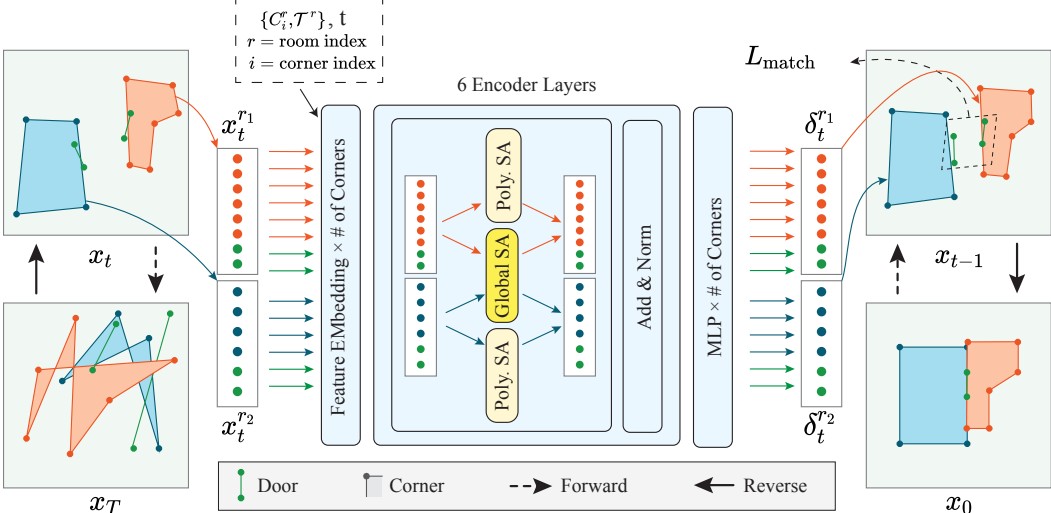

Figure 3: Illustration of our diffusion model architecture employed for the RLA task. Given the arrangement estimation $x_t = \{x_{i,t}^r\}$, the reverse process infers the noise $\{\delta_{i,t}^r\}$ and recovers $x_{t-1} = \{x_{i,t-1}^r\}$, while injecting the original room shapes $\{C_i^r\}$ and types $\{\mathcal{T}^r\}$ as the condition. Each room corner holds the room position and rotation estimation. The reverse process starts from $x_T$ and denoises towards $x_0$.

There are a few reasons. Our reverse process is based on a Transformer architecture where each position/rotation estimation becomes a node. Our approach 1) enriches the capacity of the arrangement representation (also an adaptive capacity, where a complex polygon with more corners is given more capacity); 2) allows direct communications between corners/doors for which we will have a specific loss; and 3) makes it straightforward to combine with the condition (*i.e.*, original corner coordinates and room types).

Concretely, we use $x_{i,t}^r$ to denote the position/rotation of the $r$th room/piece stored at the $i$th corner at time $t$ of the diffusion process, where $t$ varies from 0 to 1,000 in our experiments:

$$x_{i,t}^r \;=\; \left(p_{i,t}^r, o_{i,t}^r\right). \tag{1}$$

$p_{i,t}^r$ and $o_{i,t}^r$ denote the polygon-center position (a 2D vector) and the rotation. We consider rotation as a 2D vector obtained from rotation matrix including $\cos(o_{i,t}^r)$ and $-\sin(o_{i,t}^r)$. The forward process adds a noise by sampling $\delta_{i,t}^r \in \mathcal{N}(\mathbf{0}, \mathbf{I})$ with a standard cosine noise scheduling with variance $(1 - \alpha_t)$ Nichol and Dhariwal (2021b):

$$x_{i,t}^r = \sqrt{\bar{\alpha}_t}x_{i,0}^r + \sqrt{1 - \bar{\alpha}_t}\delta_{i,t}^r, \quad \bar{\alpha}_t = \frac{f(t)}{f(0)}, \quad f(t) = \cos\left(\frac{t/T + 0.008}{1 + 0.008} \cdot \frac{\pi}{2}\right)^2 \tag{2}$$

### 4.2 Reverse process

Figure 3 illustrates our reverse process for room layout arrangement, which takes the arrangement $\{x_{i,t}^r\}$ at time $t$ and infers the noise $\tilde{\delta}_{i,t}^r$ under the condition of the original room shapes $\{C_i^r\}$ (*i.e.*, a corner position with respect to the room center) and the room types as a one-hot vector $\{\mathcal{T}^r\}$.

**Feature embedding**: The reverse process is based on a Transformer architecture where every corner is a node. We initialize its feature embedding as

$$\hat{x}_{i,t}^r \leftarrow \mathrm{Lin}(x_{i,t}^r) + \mathrm{Lin}([C_i^r, r, i]) + \mathrm{MLP}(t) + \mathrm{Lin}(\mathcal{T}^r). \tag{3}$$

The first term uses a linear layer to convert a 4D vector (*i.e.*, 2 for the room center coordinate and 2 for the rotation vector) to a 256D embedding vector. The second term also uses a linear layer to convert a condition vector (*i.e.*, 2 for the corner coordinate respecting to the center of the polygon

in addition to 32 for polygon index which shows which polygon each corner belongs $r$ and 32 to show the corner index in each polygon $i$), The third term uses a 2-layer MLP to convert a time step $t$ to a 256D vector. The fourth term is extra information such as room/door type (20 types) for RLA which we extract it by applying a linear layer on 20D one-hot vector or 128D for image features in pictorial CJP)

**Attention modules**: Feature embeddings $\{\hat{x}_{i,t}^r\}$ go through six blocks of self-attention modules that have two different attention mechanisms: Polygon Self Attention (P-SA) and Global Self Attention (G-SA). P-SA limits pairwise interactions between corners in each polygon. P-SA is akin to a sparse self attention family Guo et al. (2019); Child et al. (2019); Li et al. (2019), which helps to generate consistent positions and rotations at different corners of a polygon. G-SA is a standard self-attention between all corners of a puzzle or a house. After the attention blocks, a linear layer converts 256D embedding back to a 4D representation $\tilde{\delta}_{i,t}^r$, which is used for the following denoising formula Ho et al. (2020):

$$x_{i,t-1}^r = \frac{1}{\sqrt{\alpha_t}}\left(x_{i,t}^r - \frac{1-\alpha_t}{\sqrt{1-\bar{\alpha}_t}}\tilde{\delta}_{i,t}^r\right) + \sqrt{1-\alpha_t}z. \tag{4}$$

$z \sim \mathcal{N}(0, I)$ for $t > 1$ and otherwise 0. For the final result at time $t = 0$, we take the average polygon center position and the polygon rotation.

**Loss functions**: There are two loss functions. We first follow Ho et al. (2020); Dhariwal and Nichol (2021) and use a standard noise regression loss on $\delta$:

$$L_{\text{simple}} = E_{t,x_{i,0}^r,\delta_{i,t}^r}\left[\left\|\delta_{i,t}^r - \tilde{\delta}_{i,t}^r\right\|^2\right]. \tag{5}$$

To enhance the quality of supervision, we propose a "matching" loss, specifically aimed at the vertices where incident edges meet. These vertices are shared by two polygons. We denote the indices of these two polygons as $r_1$ and $r_2$, then the corresponding vertex indices within $r_1$ and $r_2$ as $i_1$ and $i_2$, respectively. The corresponding vertices must be at the same position, and as such, we calculate the loss as the Euclidean distance between their coordinates:

$$L_{\text{match}} = E_{t,x_{i,0}^r,\delta_{i,t}^r}\left[\left\|\hat{C}_{i_1}^{r_1} - \hat{C}_{i_2}^{r_2}\right\|^2\right], \tag{6}$$

$$\hat{C}_i^r = \tilde{p}_{i,0}^r + R_{\tilde{o}_{i,0}^r}C_i^r, \tag{7}$$

$$(\tilde{p}_{i,0}^r, \tilde{o}_{i,0}^r) = \tilde{x}_{i,0}^r = \left(x_{i,t}^r - \sqrt{1-\bar{\alpha}_t}\tilde{\delta}_{i,t}^r\right)/\sqrt{\bar{\alpha}_t}. \tag{8}$$

$R_{\tilde{o}_{i,0}^r}$ denotes the rotation matrix corresponding to $\tilde{o}_{i,0}^r$. For the room layout arrangement task, we add the loss only to doors without walls, which yields superior results in our experiments. The total loss is defined as the sum of the above loss functions, $L_{\text{total}} = L_{\text{simple}} + L_{\text{match}}$. In practice, we found that adding $L_{\text{match}}$ only for $t < 500$ results in better performance.

## 5 Experimental Results and Discussions

We have implemented the system with PyTorch Paszke et al. (2019), using a workstation with a 3.70GHz Intel i9-10900X CPU (20 cores) and two NVIDIA RTX A6000 GPUs. We use the AdamW Loshchilov and Hutter (2017); Kingma and Ba (2014) optimizer with $\beta_1 = 0.9$, $\beta_2 = 0.999$, weight decay equal to 0.05, and the batch size of 512. The learning rate is initialized to 0.0005. It takes roughly 24 hours to train a model and 3 seconds to estimate the arrangement for one sample.

### 5.1 Preprocessing and Datasets

We have carefully pre-processed and prepared datasets for fair evaluation, whose details are referred to supplementary. We here provide summary points for the three tasks (See Fig. 2).

**Cross-cut Jigsaw Puzzle (CJP)**: We have used code provided by Harel *et al.* Harel and Ben-Shahar (2021) to generate 100k/1k puzzles for training/testing. The code generates a convex polygon and

cuts it by 3 to 5 lines. For the pictorial version, we have used images from COCO 2017 dataset Lin et al. (2014), that is, randomly selecting a training/testing image of the dataset for each training/testing CJP sample. Following Harel and Ben-Shahar (2021), to simulate real-world scenario, we perturb corner coordinates with noise with three different levels to create datasets under three different noise levels with thresholds equal to 0, 2, and 4.

**Voronoi Jigsaw Puzzle (VJP)**: We have generated 200k/1k puzzles for training/testing by randomly sampling 3 to 15 points inside a random rectangle and extracting their Voronoi cells as the pieces. We have created datasets with three different levels of noise. For each corner, we added a random number from a Gaussian distribution with $\sigma^2$ equal to 0, 2, and 4 for noise levels 0, 1, and 2, respectively.

**Room Layout Arrangement (RLA)**: MagicPlan (https://www.magicplan.app), a mobile software company for real estate and construction, agrees to share production data with us and the research community, where the paper introduces the MagicPlan dataset, containing room shapes and their ground-truth arrangement for 98,780 single-story houses/apartments. We split the data into 93,780/5,000 training/testing samples. MagicPlan software reconstructs room shapes one by one by asking users to click room corners through an augmented reality app. Room shapes are Manhattan-rectified as an enforcement of the app, then manually arranged to form a floorplan, which we seek to automate. Each room is associated with a room type. The number of rooms (resp. corners) in a house ranges from 3 to 10 (resp. 12 to 182). We also use RPLAN Wu et al. (2019) for evaluation, containing 60k floorplans. We divide into 55k/5k training/testing, where the number of rooms (resp. corners) in a house ranges from 3 to 8 (resp. 14 to 98). Note that RICOH dataset Saharia et al. (2021) and ZIND dataset Lambert et al. (2022b) are too small for network training and are not used in our experiments.

## 5.2 Competing methods

Harel *et al.* Harel and Ben-Shahar (2021) and Shabani *et al.* Shabani et al. (2021) are state-of-the-art methods for CJP and RLA, respectively. We have used their public implementations to compare in the corresponding tasks. Note that Harel *et al.* is not applicable to VJP or RLA, where neighboring angles may not add to $180°$ or corners may not meet. Shabani *et al.* is not applicable to CJP or BJP due to its poor scalability (i.e., exponential in the number of pieces). We have also prepared a third method based on the transformer network to be compared for RLA. The following provides more information, while the full details are in supplementary.

**Harel *et al.*** Harel and Ben-Shahar (2021) proposed a two-step algorithm for CJP: Enumerating pairs of compatible pieces by heuristics based on edge lengths or corner angles, then globally solving for the whole arrangement by a spring system.

**Shabani *et al.*** Shabani et al. (2021) enumerates arrangement candidates by heuristics and learns to regress the realism of an arrangement candidate by deep neural networks. Since the number of room/door types differs in our datasets, we made minor modifications to the data loader and the network architecture.

**TransVector** is a baseline Transformer network with a vector representation that directly estimates the pose parameters instead of iterative denoising. TransVector shares the same architecture as our denoising network at the core with the following changes: 1) Remove time-dependent features ($x_{i,t}^r$ and $t$) from the embedding (3); 2) Change the supervision $\delta_{i,t}^r$ from the noise to the position/rotation parameters; and We have also compared with the Transformer network with a raster representation, which is presented in the supplementary.

## 5.3 Main results

Table 1 compares the proposed approach against Shabani *et al.* Shabani et al. (2021) and TransVector. Shabani *et al.* runs exponentially in the number of pieces (i.e., rooms), taking hours or even days to process a single house with seven or more rooms. Therefore, we collect small houses (*i.e.*, at most six rooms) to create "Small RPLAN" and "Small MagicPlan" datasets for its evaluations. For each experimental setting (*e.g.*, Small MagicPlan), we train a network for each method.

Table 1: RLA quantitative results with two metrics: Positional Error (MPE) and Graph Editing Distance (GED). Small RPLAN (resp. Small MagicPlan) is a subset of the corresponding full dataset, consisting of houses with at most 6 rooms. The small datasets are created for Shabani *et al*. Shabani et al. (2021), which is not scalable to many rooms.

| Dataset | Small RPLAN | | Full RPLAN | | Small MagicPlan | | Full MagicPlan | |
|---|---|---|---|---|---|---|---|---|
| Metric | MPE ($\downarrow$) | GED ($\downarrow$) | MPE ($\downarrow$) | GED ($\downarrow$) | MPE ($\downarrow$) | GED ($\downarrow$) | MPE ($\downarrow$) | GED ($\downarrow$) |
| Shabani *et al*. | 29.44 | 1.28 | ✗ | ✗ | 36.63 | **1.89** | ✗ | ✗ |
| TransVector | 36.09 | 1.51 | 46.18 | 2.27 | 40.80 | 2.38 | 53.11 | 6.41 |
| Ours | **8.65** | **0.90** | **10.55** | **0.97** | **32.76** | 1.95 | **40.81** | **3.09** |

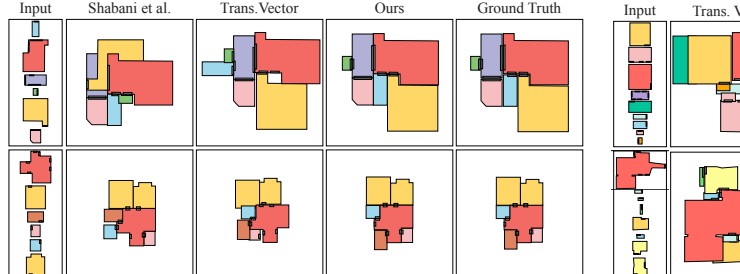

Figure 4: RLA arrangement results with Small Magic-Plan (top) and Small RPLAN (bottom).

Figure 5: RLA arrangement results with Full MagicPlan dataset.

Our method outperforms all the others in all the settings and metrics, except the GED metric for Small MagicPlan, where the existing state-of-the-art Shabani *et al*. achieves a slightly better score. Their method enumerates all possible candidate arrangements by matching doors and evaluates the realism of each arrangement one by one. Their run-time is exponential in the number of rooms, and the system involves many heuristics. TransVector is end-to-end and achieves comparable performance with Shabani *et al*.; in fact, a much better MPE score for Small RPLAN. However, our system performs much better in every metric, demonstrating the power of Diffusion Models even for non-generative tasks, in our case, room layout arrangement. Figure 4 compares Shabani *et al*., Transvector and ours again on the small datasets. Figure 11 shows our results for the most challenging setting (*i.e.*, Full MagicPlan dataset) comparing to Transformer with vector representation.

Table 2 presents a comparative analysis for CJP and VJP at three different noise levels. We have trained our model only for a noise-free case (i.e., 0 noise level) and used for the other two levels. Our approach consistently outperforms the current state-of-the-art. The performance gap is significant with the presence of noise, because Harel *et al*. relies on critical assumptions that matching edges have the same distance and neighboring angles sum to $180°$. They have some tolerance to cope with noisy inputs but do not do well against our learning-based approach. Figs. 6, 7, and 8 show sample arrangement results for apictorial CJP, pictorial CJP, and VJP, respectively, supporting the above quantitative results. Note that the table shows the numbers for apictorial CJP. Due to the time complexity of Harel *et al*., we have only calculated the metrics on 20 samples for pictorial CJP, showing the same trend in results. The full details are referred to the supplementary, which also contains experiments on duplicate or missing pieces, where our method is surprisingly robust.

### 5.4 Ablation studies

We choose the room layout arrangement task for further ablation studies on our method. Table 7 shows the contributions of our two attention mechanisms (P-SA, G-SA) and the door matching loss. G-SA provides communications between every pair of corners in a house, and its removal has the most impact on the performance. Removing P-SA or the matching loss also leads to a significant drop in both MPE and GED metrics. Our method with all the components achieves the highest performance.

Table 2: CJP and VJP quantitative results with three metrics. Our method consistently shows superior performance with significant margins under the presence of noise. The second row shows ours without the matching loss. Harel *et al.* is unable to handle VJP due to the assumptions (See 5.2).

| Dataset | Cross-cut Jigsaw Puzzle | | | | | | | | | Voronoi Jigsaw Puzzle | | | | | | | | |
|---|---|---|---|---|---|---|---|---|---|---|---|---|---|---|---|---|---|---|
| Metric | Overlap (↑) | | | Precision (↑) | | | Recall (↑) | | | Overlap (↑) | | | Precision (↑) | | | Recall (↑) | | |
| Noise Level | 0 | 1 | 2 | 0 | 1 | 2 | 0 | 1 | 2 | 0 | 1 | 2 | 0 | 1 | 2 | 0 | 1 | 2 |
| Harel *et al.* | 0.91 | 0.70 | 0.30 | 0.95 | 0.77 | 0.33 | **0.99** | 0.78 | 0.30 | ✗ | ✗ | ✗ | ✗ | ✗ | ✗ | ✗ | ✗ | ✗ |
| Ours (w/o $L_{match}$) | 0.91 | 0.90 | 0.89 | 0.94 | 0.92 | 0.90 | 0.77 | 0.76 | 0.73 | 0.65 | 0.64 | 0.63 | 0.75 | 0.71 | 0.71 | 0.53 | 0.52 | 0.52 |
| Ours | **0.94** | **0.94** | **0.93** | **0.97** | **0.95** | **0.93** | 0.91 | **0.92** | **0.91** | **0.70** | **0.68** | **0.67** | **0.78** | **0.75** | **0.74** | **0.60** | **0.57** | **0.55** |

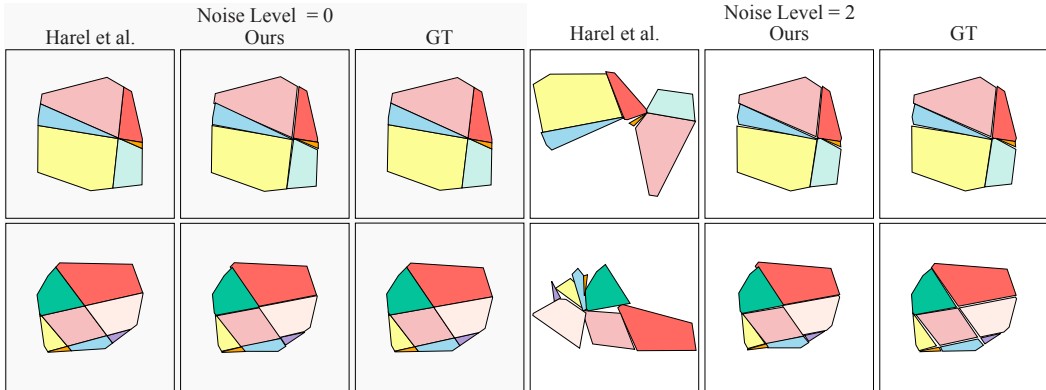

Figure 6: Apictorial CJP arrangement results. Both methods work well in a noise-free case (left), but only our method maintains the performance with the presence of noise (right).

Table 3: Contributions of our two attention mechanisms (P-SA, G-SA) and the door matching loss ($L_{match}$). Full MagicPlan is used. ✓indicates the feature being used. In case of $L_{match}$ "Doors" means matching loss has been applied only on door corners and "All corners" means matching loss has been applied to all corners including door corners.

| P-SA | G-SA | $L_{match}$ | MPE (↓) | GED (↓) |
|---|---|---|---|---|
| None | ✓ | None | 48.2 | 4.9 |
| ✓ | ✓ | None | 43.4 | 4.5 |
| ✓ | None | All corners | 55.3 | 9.4 |
| None | ✓ | All Corners | 45.1 | 4.6 |
| ✓ | ✓ | All corners | 41.8 | 3.6 |
| ✓ | None | Doors | 56.9 | 9.3 |
| None | ✓ | Doors | 45.2 | 4.3 |
| ✓ | ✓ | Doors | 40.8 | 3.1 |

Table 4: Effects of the room-type (R-type) and the Door information. Full MagicPlan is used. ✓indicates the information being used. When a room-type is not used, we set a zero vector as a room-type one-hot vector, when room type is noisy we assign each room with a random room type . When the door information is not used, we do not pass the door-corner nodes to the network.

| Train | | Test | | MPE(↓) | GED (↓) |
|---|---|---|---|---|---|
| R-Type | Door | R-Type | Door | | |
| ✓ | ✓ | None | ✓ | 48.4 | 3.8 |
| None | ✓ | None | ✓ | 47.4 | 3.6 |
| ✓ | ✓ | Noisy | ✓ | 49.7 | 4.0 |
| Noisy | ✓ | Noisy | ✓ | 49.3 | 3.9 |
| ✓ | ✓ | ✓ | None | 46.9 | 5.2 |
| ✓ | None | ✓ | None | 45.5 | 5.2 |
| ✓ | ✓ | ✓ | ✓ | 40.8 | 3.1 |

Our layout arrangement uses a redundant representation (i.e., all the corners store the position and the orientation estimates of a piece), enriching the capacity and enabling direct communications between room/door corners (Sect. 4.1). To assess the effects of this redundancy, we have created a variant of our system with a compact representation, that is, each room/door has only a single node estimating a single copy of the room position and the orientation. We aggregate corner coordinates into a single embedding vector and pass as a condition (See supplementary for the details). The MPE/GED metrics for Full MagicPlan change from (41.23/3.16) to (51.62/5.52), a significant performance drop showing the importance of our redundant representation.

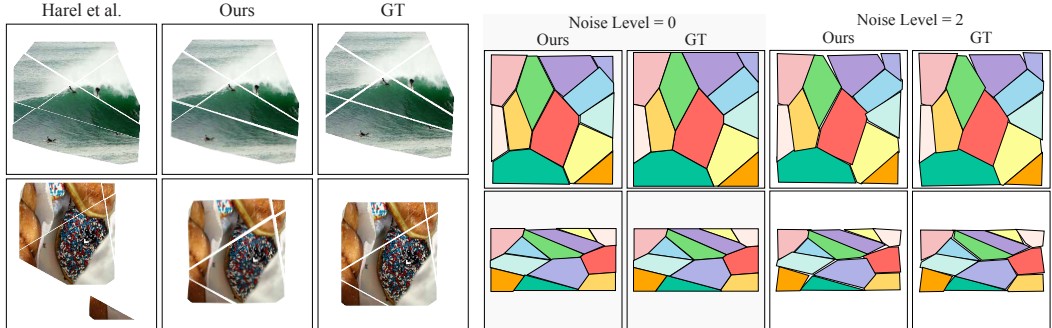

Figure 7: Pictorial CJP arrangement results.    Figure 8: VJP arrangement results.

A popular approach in the RLA literature is to align door detections/annotations to enumerate arrangement candidates Lambert et al. (2022a); Shabani et al. (2021). Our approach directly learns to infer the arrangement without relying on such "hard" heuristics. To further demonstrate the power of our approach, we remove or alter the room type and the door information during training and testing, where the current state-of-the-art methods Lambert et al. (2022a); Shabani et al. (2021) completely fail. Table 8 shows some performance drop, but the effects are marginal. Our numbers are still much better than TransVector with the full information (MPE=53.11 and GED=6.41 in Table 1), the only competing method capable of handling this most challenging setting. To evaluate the capacity of our network to handle overlaps, we conducted an experiment. In this experiment, we augmented our train/test dataset by randomly selecting up to two rooms per house. For each selected room, we employed one of the following strategies: 1) Duplicating the room with the ground truth (GT) room types. 2) Duplicating the room with random room types. 3) Enlarging the room by 20% to create partial overlaps. The dataset used for this experiment is Full MagicPlan dataset. Our evaluation metrics, MPE/GED, resulted in 48.3/3.9, respectively. Please also see the supplementary document and the video for more results, more visualizations, and animations of the denoising process.

## 5.5 Conclusion

This paper introduced an end-to-end neural architecture for spatial puzzle solving tasks. The proposed approach is faster, robust to data corruptions, end-to-end, and far superior to existing methods in all the metrics in the variety of tasks, namely Cross-cut jigsaw puzzle (pictorial and apictorial), Voronoi jigsaw puzzle, and room layout arrangement. Despite numerous advantages, our approach faces certain limitations. The primary challenge is its substantial need for training data, which limits our ability to process smaller datasets. For the room layout arrangement task, the utilization of raw image information could further enhance performance as discussed in Shabani et al. (2021); Lambert et al. (2022a). However, this would significantly increase the amount of data transfer from a mobile device, where on-device processing would become desirable. One key future work is the development of data-efficient (at training) and computation-efficient (at testing) neural architectures Kitaev et al. (2020). To our knowledge, this paper is the first to demonstrate that Diffusion Models, generally regarded as powerful generative models, are also effective in solving various challenging spatial arrangement tasks. The paper has the potential to motivate other researchers to further expand the applicability of emerging Diffusion Models, moving beyond content generation and into a myriad of other tasks.

**Acknowledgment**: This research is partially supported by NSERC Discovery Grants with Accelerator Supplements and the DND/NSERC Discovery Grant Supplement, NSERC Alliance Grants, and the John R. Evans Leaders Fund (JELF). We are also thankful to Magicplan for sharing the datasets.

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

# A Methods details of our system and competing methods

We benefit from Transformers in our task in two ways. First, Transformers provide the capability of processing sequences with different lengths, which we use to process different number of room layouts/corners in the houses. Second, we utilize the self-attention module of Transformers to create optimal interaction and information-sharing among input tokens. These two features make Transformers an ideal backbone for our model. Our method uses six Transformer encoder blocks, and attention in each block has four heads. We also use an MLP For converting 256D Transformer output to rotation and position (4D). To keep the experiments fair, we use the same architecture for our transformer baselines as much as possible. In the following, we provide details corresponding to each of the baselines.

**Transformer with a raster representation** (TransRaster) uses the raster images to represent the input room layouts/types and the output room positions. Note that this baseline does not handle rotations as explained below. An input room layout is represented as a 20-channel $256 \times 256$ semantic segmentation image, where there are 20 room/door types. The room center is aligned with the center of an image. An output room position is represented as a $256 \times 256$ room occupancy image, which is ideally a translated version of the input room segmentation image at the correct room location. Given an output room occupancy image, we perform an exhaustive search over the possible room translations and find one with the most overlap between the occupancy image and the translated room segmentation image. [3] We use VisionTransformer Dosovitskiy et al. (2020) with a CNN decoder that takes a set of input room segmentation images and produces a set of room occupancy images.

In the other word TransRaster uses an encoder part of U-Net, which has 8 down-sampling blocks, converting each input room layout to a feature map of dimension 512. Each feature map (corresponding to a room layout) will become one input token for the Transformer. Input sequences length is equal to the number of rooms in a house, and information is shared among different rooms. We use six Transformer encoder blocks, and attention in each block has four heads. We pass the output of Transformer to a U-Net up-sampling model with eight Up-sampling blocks to change the dimension from 512 to $256 \times 256$.

**Transformer with a vector representation** uses the same backbone as our method; a linear layer converts the 28D input vector (i.e., 2 for the original corner coordinate and 20 for the room/door type one-hot vector) to the 256D feature map, six Transformer encoder blocks, and the attention in each block have four heads. We also use an MLP to convert 256D output embedding to 4D output.

**Diffusion model, one room per node** encodes each node as corresponding to a room instead of a corner in the room. To ease the implementation, we set the maximum number of nodes per room to 20 and we pad extra nodes when the room has less than 20 nodes with 0. We flatten the conditions per room and then use a linear layer to convert it to a 256D embedding vector. Each feature map represents a room and an input token for Transformer, we use the same Transformer as our method. After the Transformer blocks, a linear layer converts 256D output to 4D (i.e., 2 for the position and 2 for the rotation).

**Shabani *et al*.** Shabani et al. (2021) takes the input layout of each room with the resolution of $256 \times 256$ with the same number of channels as the number of room types to pass each pixel as a one-hot vector of the corresponding room type. We use the same model as Shabani et al. (2021) and change the number of input channels to 11 for RPLAN and 20 for MagicPlan. To generate the arrangement candidates, we use the given room layouts of our dataset to connect doors, while we also use overlap filtering to reduce the number of candidates. Note that our datasets is significantly larger than the one in Shabani et al. (2021), enabling us to randomly select a positive or a negative candidate in each iteration and therefore remove the class imbalance weight used in Shabani et al. (2021). During the training for each house, we randomly select a GT with the label 1 or a faulty candidate with the label [0, 1) based on the number of mismatched doors. During the test, we pass all the possible candidates of each house and select the candidate with the highest score as the final prediction.

---

[3] We could expand the search space with possible room rotations, but rooms are often symmetric. To be simple, we use this baseline only for experiments when ground-truth rotations are given.

**Harel *et al*.** Harel and Ben-Shahar (2021) proposed a two-step algorithm for CJP. Their method considers two types of constraints to find plausible mates based on the length and angle of different pairs of connections. By estimating the matings hierarchically using these constraints, they approach the problem of finding positions as a multi-body spring-mass system. We utilize the authors' provided implementation[4] for comparison with our method. With a test dataset of 1000 crossing-cut puzzles, we restrict the running time of the spring-system algorithm to 2 minutes per puzzle. Furthermore, unlike the provided implementation, we also consider failure cases in the metrics. Regarding the pictorial case, the authors score a candidate mating by extrapolating the images of puzzle pieces and considering the difference of the mean color value on the edges. We do not impose any time limit as we evaluate only on 20 samples.

## B   Datasets Details and Preprocessing

| Room Type | 3 | 4 | 5 | 6 | 7 | 8 | 9 | 10 | All |
|---|---|---|---|---|---|---|---|---|---|
| Master bedroom | 0.20 | 0.26 | 0.29 | 0.32 | 0.39 | 0.49 | 0.49 | 0.53 | 0.34 |
| Living room | 0.52 | 0.56 | 0.59 | 0.65 | 0.71 | 0.75 | 0.79 | 0.79 | 0.65 |
| Kitchen | 0.42 | 0.47 | 0.52 | 0.59 | 0.68 | 0.71 | 0.76 | 0.79 | 0.59 |
| Bathroom | 0.54 | 0.70 | 0.85 | 0.96 | 1.12 | 1.28 | 1.33 | 1.47 | 0.96 |
| Toilet | 0.07 | 0.11 | 0.15 | 0.22 | 0.22 | 0.25 | 0.27 | 0.26 | 0.18 |
| Corridor | 0.07 | 0.12 | 0.15 | 0.19 | 0.25 | 0.32 | 0.37 | 0.45 | 0.21 |
| Closet | 0.13 | 0.18 | 0.22 | 0.32 | 0.48 | 0.68 | 0.92 | 1.22 | 0.41 |
| Hall | 0.35 | 0.55 | 0.68 | 0.77 | 0.85 | 0.91 | 0.98 | 1.08 | 0.73 |
| Laundry room | 0.05 | 0.05 | 0.05 | 0.07 | 0.10 | 0.13 | 0.18 | 0.23 | 0.09 |
| Bedroom | 0.34 | 0.69 | 1.08 | 1.32 | 1.51 | 1.74 | 1.93 | 2.10 | 1.23 |
| Balcony | 0.05 | 0.08 | 0.16 | 0.32 | 0.40 | 0.48 | 0.56 | 0.64 | 0.29 |
| Dining room | 0.13 | 0.12 | 0.12 | 0.14 | 0.17 | 0.19 | 0.22 | 0.25 | 0.15 |
| Private office | 0.00 | 0.01 | 0.02 | 0.05 | 0.05 | 0.07 | 0.08 | 0.10 | 0.05 |
| Den | 0.05 | 0.05 | 0.06 | 0.7 | 0.09 | 0.11 | 0.12 | 0.12 | 0.08 |
| Storage | 0.00 | 0.01 | 0.01 | 0.02 | 0.02 | 0.02 | 0.03 | 0.03 | 0.02 |
| Others | 0.00 | 0.01 | 0.01 | 0.02 | 0.03 | 0.04 | 0.04 | 0.07 | 0.03 |
| Doors | 2.84 | 3.82 | 4.82 | 5.43 | 6.92 | 8.02 | 6.10 | 10.18 | 5.90 |

Table 5: MagicPlan dataset consists of floorplans with 3 to 10 rooms. The table shows average number of rooms with a specific room type based on the total number of rooms in the house.

We normalize the puzzles/floorplans for each task and dataset by scaling them to fit within a $1 \times 1$ square, and we also resize all corresponding images to dimensions of $256 \times 256$ in the case of pictorial CJP. While this normalization process does not introduce any essential additional information during testing in CJP and VJP, it could potentially enables the network to cheat in RLA, as the longer extent of arranged floorplans is always fixed to $1$. To address this issue, during testing in RLA, we apply a random scaling factor in the range of [0.8, 1.0] to the room shapes of each house. In the subsequent sections, we provide a detailed description of each dataset. In the following, we provide additional statistics for our floorplan datasets.

The Voronoi Jigsaw Puzzle dataset consists of 200k training puzzles and 1k testing puzzles. These puzzles were created by randomly selecting 3 to 15 points, The individual pieces of the puzzles were obtained by extracting the Voronoi cells corresponding to these points. There are 1,066, 14,033, 23,715, 20,279, 16,073, 15,235, 16,428, 15,018, 16,318, 15,096, 16,487, 16,233, and 15,670 puzzles with 3, 4, 5, 6, 7, 8, 9, 10, 11, 12, 13, 14, and 15 pieces respectively. Each piece has a minimum, maximum, and average of 3, 20, and 4.51 corners respectively. The minimum, maximum, and average number of corners per puzzle are 10, 93, and 42.24.

Cross-cut Jigsaw Puzzle (CJP) are consist of 100k training and 1k testing puzzles, where each one were generated using  Harel and Ben-Shahar (2021) method which generate a convex polygon and cuts it by 3 to 5 lines. There are 1719, 6046, 15854, 14521, 6905, 10929, 12065, 8361, 6521, 8192, 7663, 4642, 1508, 73, and 1 puzzles with 3, to 18 pieces respectively. Each piece has a minimum,

---

[4]https://icvl.cs.bgu.ac.il/polygonal-puzzle-solving/

maximum, and average of 3, 13, and 4.47 corners respectively. The minimum, maximum, and average number of corners per puzzle are 16, 76, and 41.99.

MagicPlan dataset consists of roughly 98K houses/apartments, which we divide into 93K training and 5K testing samples. The number of rooms in a house ranges from 3 to 10. Concretely, 11661, 16322, 19171, 17582, 13200, 9649, 6780, and 4415 houses contain 3, 4, 5, 6, 7, 8, 9, and 10 rooms, respectively. The minimum and maximum numbers of corners in a house are 12 and 182. Table 5 shows average number of rooms with a specific room type based on the total number of rooms in the house.

In the RPLAN dataset, we divide 60K samples in RPLAN to 55K train and 5K test. The number of rooms in a house ranges from 3 to 8. Concretely 99, 582, 5083, 19551, 21921, and 13235 houses contain 3, 4, 5, 6, 7, and 8 rooms, respectively.

## C  Additional ablation studies

### C.1  Additional ablation studies on room layout arrangement

Table 6: Main quantitative results with two metrics: Positional Error (MPE) and Graph Editing Distance (GED). This table show a case where the ground-truth rotations are given, as TransRaster baseline cannot handle rotations. Small RPLAN (resp. Small MagicPlan) is a subset of the corresponding full dataset, consisting of houses with at most 6 rooms. The small datasets are created for Shabani *et al*., which is not scalable to many rooms. Our method is stochastic and shows both the mean and the standard deviation.

| Dataset | Small RPLAN | | Full RPLAN | | Small MagicPlan | | Full JigsawPlan | |
|---|---|---|---|---|---|---|---|---|
| Metric | MPE ($\downarrow$) | GED ($\downarrow$) | MPE ($\downarrow$) | GED ($\downarrow$) | MPE ($\downarrow$) | GED ($\downarrow$) | MPE ($\downarrow$) | GED ($\downarrow$) |
| Shabani *et al*. | 17.6 | 1.0 | ✗ | ✗ | 32.2 | 1.1 | ✗ | ✗ |
| TransRaster | 13.9 | 1.2 | 15.7 | 2.1 | 36.1 | 2.1 | 41.9 | 4.1 |
| TransVector | 12.9 | 1.1 | 13.9 | 2.0 | 37.7 | 1.9 | 42.8 | 4.0 |
| Ours | **4.6**±0.7 | **0.4**±0.0 | **5.4**±0.7 | **0.6**±0.0 | **17.5**±0.8 | **1.0**±0.4 | **27.9**±0.7 | **2.7**±0.5 |

Figure 13 shows the raw estimated position information at each room/door corner before the room-wise averaging. Since the ground-truth has the same pose parameters for all corners in a room/door, the network learns to produce consistent parameters. Figure 14 shows five pose estimation results by our system while varying the initial noise $x_T$. While there are minor differences, the overall room arrangements are similar and close to the ground-truth, indicating that the Diffusion model is capable of producing consistent results given enough constraints as a pose estimation system, as opposed to a generative model whose original goal is to create a diverse set of answers.

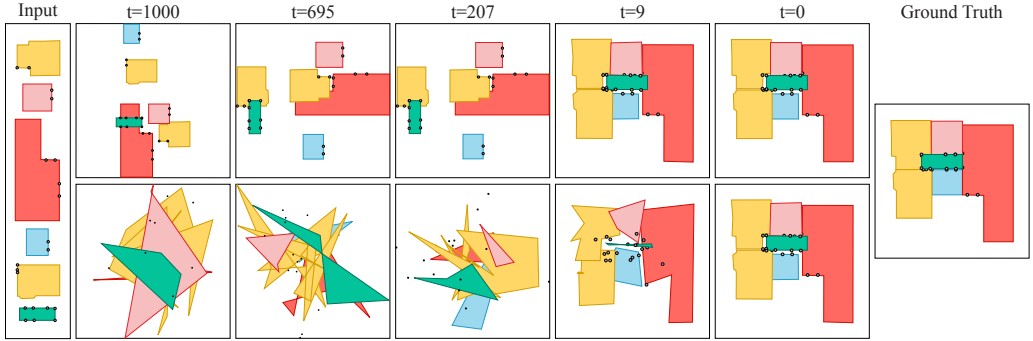

Figure 9: Visualization of predicted layouts at step "t"s. At t=1000, position parameters at each corner are initialized by a Gaussian noise, and at t=0, there is the final predicted layout. The top row shows the predicted layout without averaging/voting, and the bottom row shows with averaging/voting. To make it more clear, we show doors by their corners.

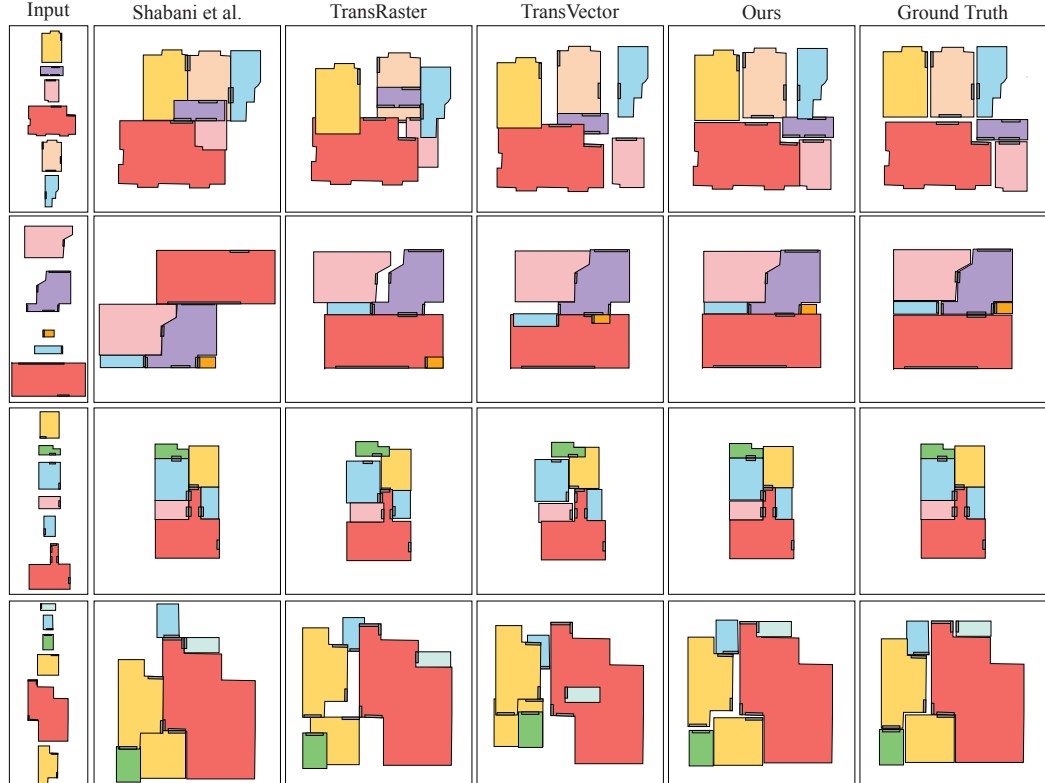

| Input | Shabani et al. | TransRaster | TransVector | Ours | Ground Truth |

Figure 10: Qualitative evaluations of our approach against the three competing methods. The top two rows are from Small MagicPlan. The bottom two rows is from Small RPLAN. The GT rotations are given for all the cases to enable comparisons with all the methods.

### C.1.1 Additional ablation on RPLAN

The main paper shows the ablation studies on the MagicPlan dataset in case of room layout arrangment task. This part of supplementary will present the same study results on RPLAN dataset. Table 7 shows the impact of our attention module and door matching loss on performance and Table 8 shows the impact of noise in the room type and door detection on our performance, although there is a performance drop, our method still works better than the competing methods.

Table 7: Co attention mechanisms (P-SA, G-SA) and the door matching loss ($L_{match}$). Full RPLAN is used. ✓indicates the feature being used. In case of $L_{match}$ "Doors" means matching loss has been applied only on door corners and "All corners" means matching loss has been applied to all corners including door corners.

| P-SA | G-SA | $L_{match}$ | MPE ($\downarrow$) | GED ($\downarrow$) |
|------|------|-------------|--------------------|--------------------|
|      | ✓    |             | 25.6               | 1.6                |
| ✓    | ✓    |             | 24.2               | 1.5                |
| ✓    |      | Doors       | 36.9               | 2.4                |
|      | ✓    | Doors       | 22.1               | 1.1                |
| ✓    | ✓    | All Corners | 10.7               | 0.9                |
| ✓    | ✓    | Doors       | 10.5               | 0.9                |

Table 8: Effects of the room-type (R-type) and the Door information. Full RPLAN is used. ✓indicates the information being used. When a room-type is not used, we set a zero vector as a room-type one-hot vector. When the door information is not used, we do not pass the door-corner nodes to the network.

| Train | | Test | | MPE($\downarrow$) | GED ($\downarrow$) |
|-------|------|--------|------|-------------------|--------------------|
| R-Type | Door | R-Type | Door | | |
| ✓     | ✓    |        | ✓    | 17.3              | 1.4                |
|       | ✓    |        | ✓    | 16.4              | 1.5                |
| ✓     | ✓    | ✓      |      | 15.1              | 1.9                |
| ✓     |      | ✓      |      | 14.3              | 1.9                |
| ✓     | ✓    | ✓      | ✓    | 10.5              | 0.9                |

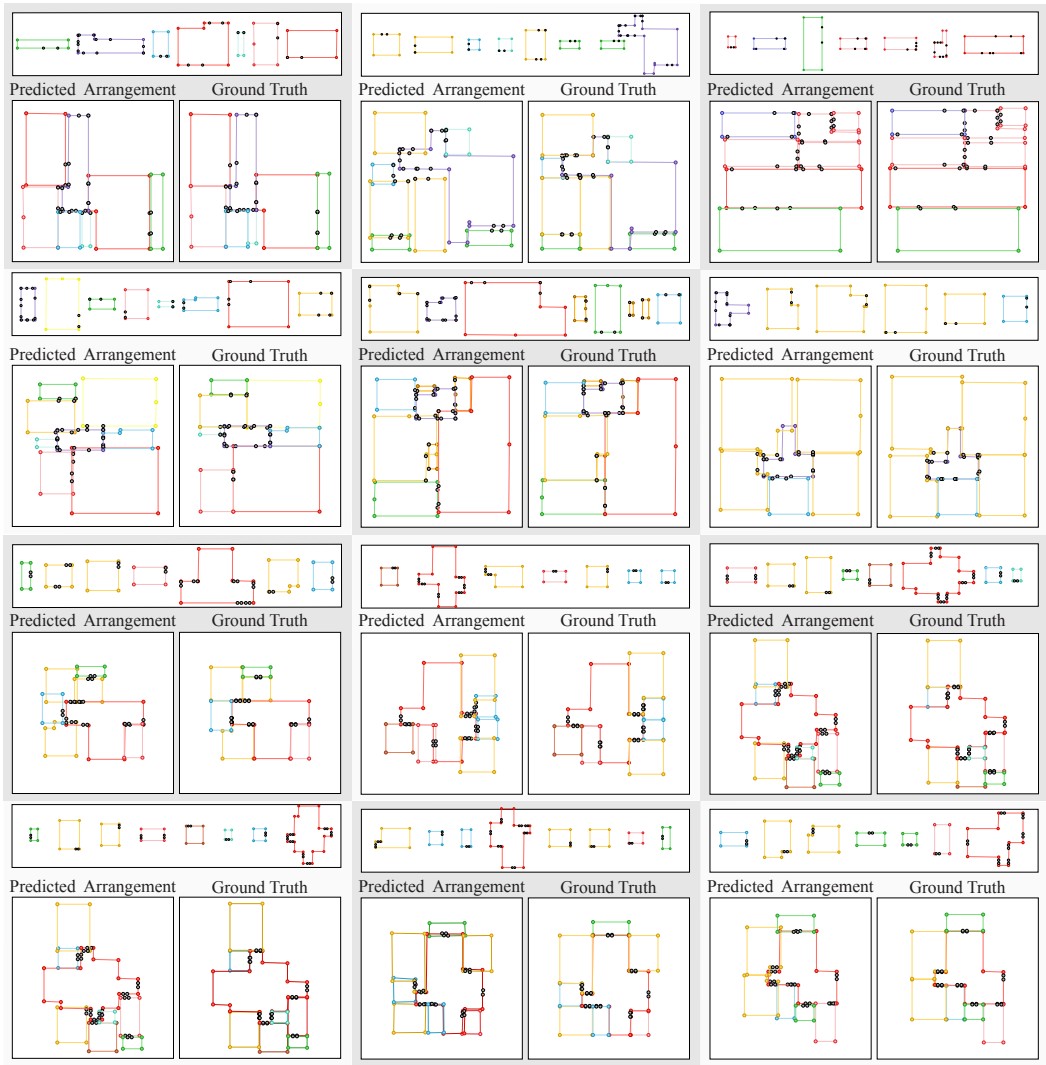

Figure 11: Qualitative evaluations of our method for Full MagicPlan dataset without GT rotations top two rows, and Full RPLAN dataset without GT rotations bottom two rows. We show edges and corners here to show overlaps and noisy annotations more clear.

## C.2 Additional ablation studies on puzzle solving

We have provided additional qualitative results of our method in Figure 15 and Figure 16 including noisy samples or samples with missing or duplicate pieces. In case of missing and duplicate experiment, we repeat (remove) each piece with a probability of $10\%$. Table 9 presents the evaluation metrics of missing and duplicate experiments.

### C.2.1 Pictorial Cross-cut Jigsaw Puzzle

To enhance the integration of image information into our pictorial puzzle diffusion models, we employed a two-step approach. Firstly, we pretrained an auto-encoder utilizing the puzzle pieces. This auto-encoder served as the image embedder for our diffusion model, enabling the conversion of each puzzle into a compact 128D feature vector.

The pretraining process involved training the model to downsample an input image of dimensions $3 \times 256 \times 256$ to a compressed representation of size $32 \times 2 \times 2$ within the encoder, and subsequently reconstructing the original image size in the decoder. We employed the mean squared error

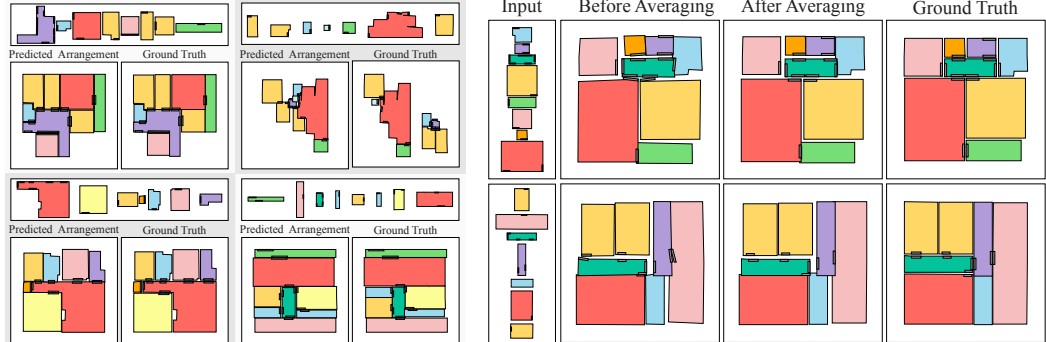

Figure 12: RLA arrangement results with Full MagicPlan dataset. On left two successful cases, on right two failed cases,. Our failures are often attributed to 1) Rare building architecture (top-right) and 2) Inherent ambiguity (bottom-right ), whose tasks are challenging even for humans.

Figure 13: The final room arrangement before and after averaging. Our diffusion model estimates a room position/rotation at each room corner, which may not be consistent in a room. The final arrangement is obtained by taking the average position and rotation within each room.

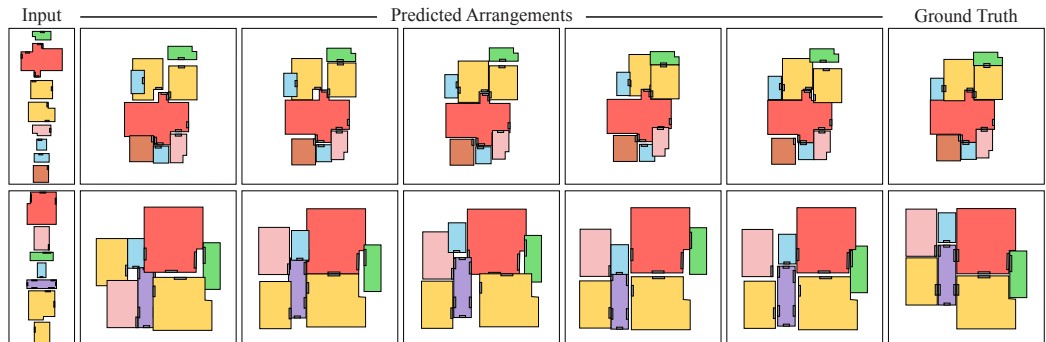

Figure 14: A diffusion model is stochastic and produces a different result every time. The middle rows show five different pose estimation results. The top (resp. bottom) is from Full RPLAN (resp. Full MagicPlan) dataset.

(MSE) loss function during training. However, to focus our model's attention on learning the texture features, given that the diffusion model already captured the geometry features, we applied the loss function exclusively to the pixels within the puzzle piece.

By adopting this selective application of the loss function, we prioritize the acquisition of texture-based details, as the geometric characteristics are already embedded within the diffusion model.

Quantitatively, we also evaluated our method on the full Cross-cut dataset to measure the effectiveness of the pictorial information compared to apictorial scenario. We found that the model converges faster when using pictorial information while it achieves slightly better overlap score of 0.9417 compared to 0.9398 in apictorial scenario. Figure 17 shows additional qualitative results of our method compared to Harel *et al*. Harel and Ben-Shahar (2021).

Table 9: Effects of the Missing or Duplicate pieces in puzzle solving problem. ✓indicates it if missing or duplicate piece were presented during test time. In training time we do not have duplicate or missing piece presented to show our model robustness to unseen noise during test.

| | | Cross-cut | | | Voronoi | | |
|---|---|---|---|---|---|---|---|
| Missing | Duplicate | Overlap ($\uparrow$) | Precision ($\uparrow$) | Recall ($\uparrow$) | Overlap ($\uparrow$) | Precision ($\uparrow$) | Recall ($\uparrow$) |
| ✓ | - | 0.88 | 0.92 | 0.82 | 0.68 | 0.68 | 0.56 |
| - | ✓ | 0.92 | 0.97 | 0.88 | 0.67 | 0.71 | 0.57 |
| - | - | 0.94 | 0.97 | 0.91 | 0.70 | 0.78 | 0.60 |

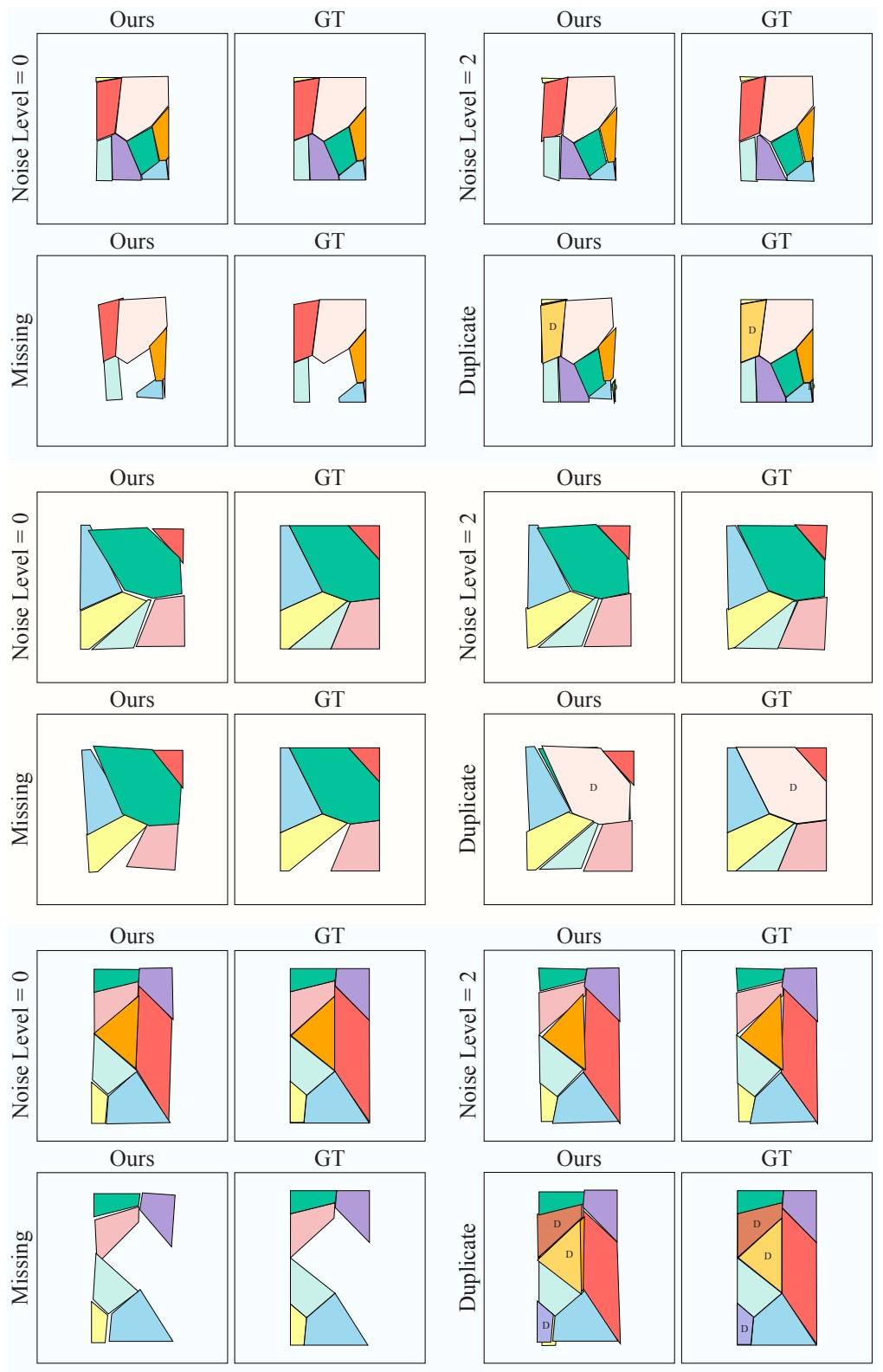

Figure 15: Additional qualitative results of Voronoi jigsaw puzzle are presented in four different setups: 1) No noise, 2) Noise level 2, 3) Missing piece, and 4) Duplicate piece (D indicates the duplicated pieces).

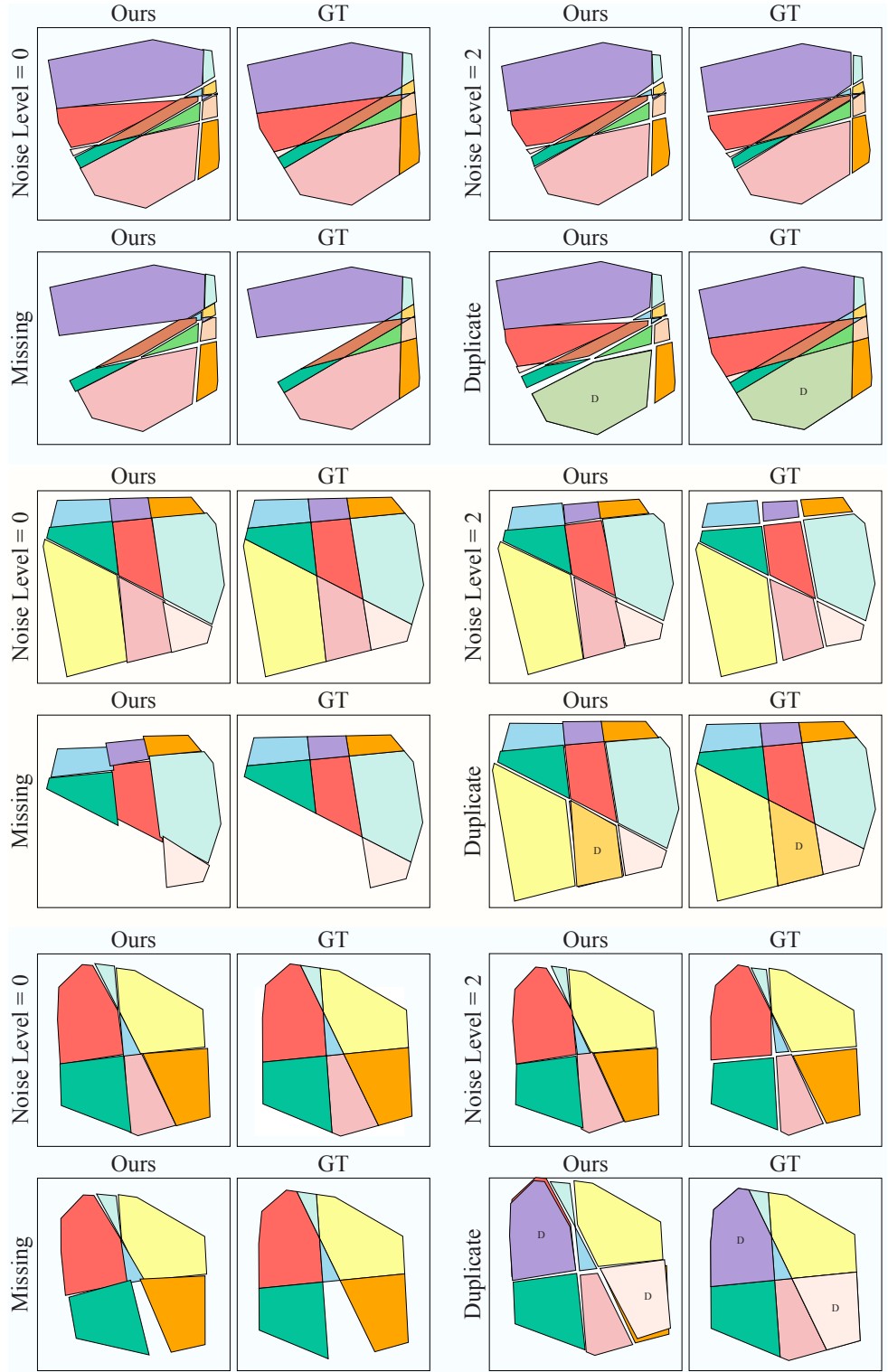

Figure 16: Additional qualitative results of Cross-cut jigsaw puzzle are presented in four different setups: 1) No noise, 2) Noise level 2, 3) Missing piece, and 4) Duplicate piece (D indicates the duplicated pieces).

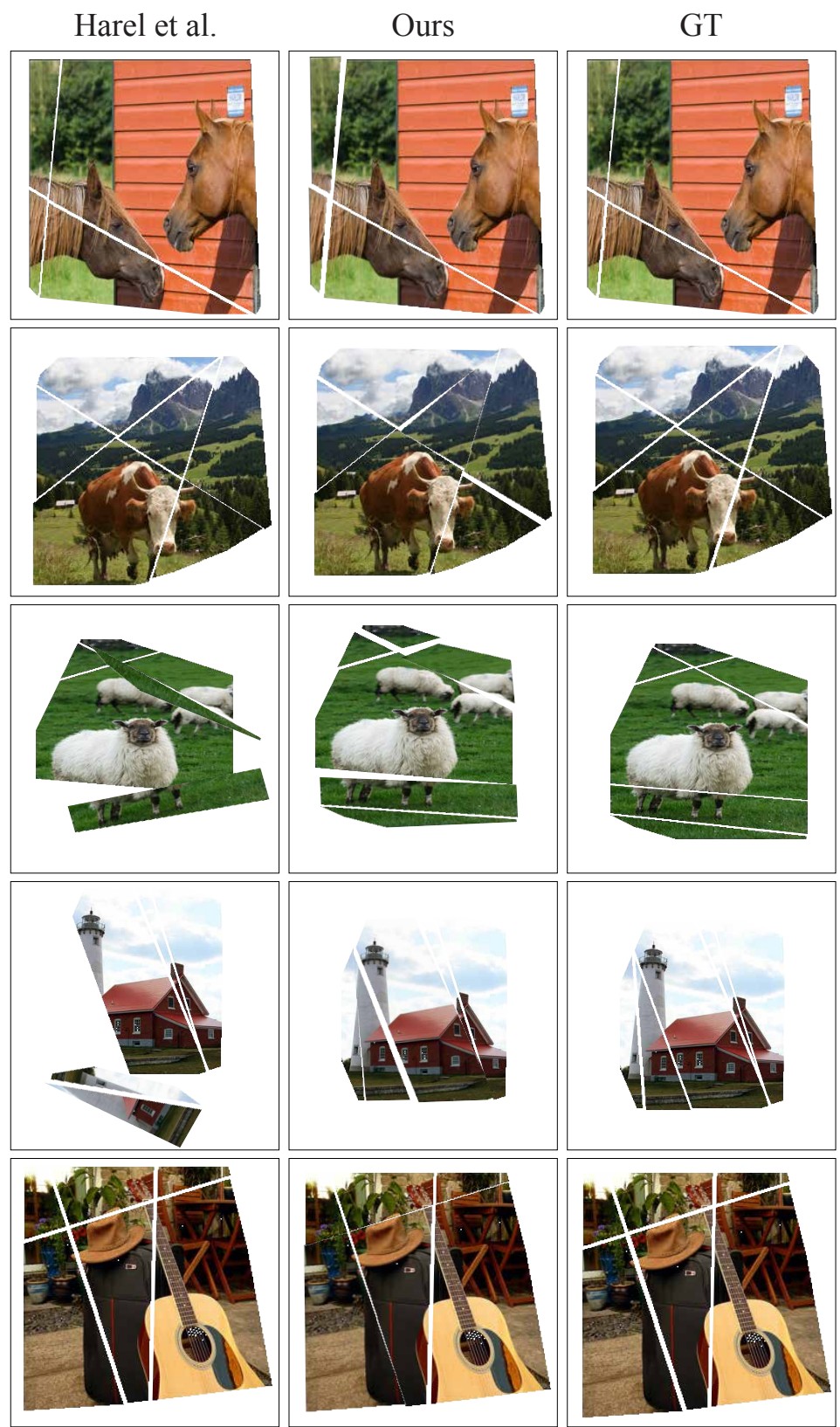

Figure 17: Additional qualitative results of pictorial Cross-cut jigsaw puzzle compared to Harel *et al*. Harel and Ben-Shahar (2021).

