# OpenReview forum: "Puzzlefusion: Unleashing the Power of Diffusion Models for Spatial Puzzle Solving"
_NeurIPS.cc/2023/Conference — NeurIPS 2023 spotlight_

### Official Review · Reviewer_zazU · 2023-07-02

**Soundness:** 3 good
**Presentation:** 3 good
**Contribution:** 2 fair
**Rating:** 5
**Confidence:** 4

**Summary:**

This paper applies conditional diffusion model to address the problem of spatial puzzle solving and show its real application on Cross-cut Jigsaw Puzzle (CJP), Voronoi Jigsaw Puzzle (VJP) and room layout arrangement (RLA). Unlike previous methods that rely on enumerating and verifying pairwise alignment and may struggle when the complexity of global arrangement increases, this approach aligns all pieces in one pass. Qualitative and quantitative evaluations show the proposed method outperforms competitive methods.

**Strengths:**

+ The writing is easy to follow.
+ The proposed method verifies the potential of the generative model on spatial puzzle solving, which may inspire to address more complex spatial puzzles.
+ Experiments with noisy spatial puzzles demonstrate the robustness of the proposed methods.

**Weaknesses:**

+ The application and the method are not properly motivated. I cannot come up with a case that we need to put together multiple floorplan regions. It looks a little weird. Besides, the other two applications, i.e., CJP and VJP, are too simple.

+ I did not see how the method solves complex cases as raised in line 30.  Full MagicPlan (RPLAN) has at most 10 (8) room patches, while Small MaginPlan (RPLAN) has at most 6 (6) room patches, thus they have minor differences in complexity. The search space of these examples is small and may not need a learning-based method.

+ Methodology: The canonical diffusion denoising probabilistic model requires quite some denoising steps, the proposed method does not apply any modifications to remedy the low time efficiency.


**Questions:**

+ Experiments:
    + It seems better to derive more analysis on ablation studies. For example, in Tab. 3 lines 5 and 8, why does matching loss applying on all corners have a slightly lower performance than the optimal choice?  Besides, feature redundancy at each corner seems critical, what is the intuition behind this design? Can it be explained by providing more training examples to the model?
    + There should be more experimental results. Does reference [21] have more competitive results on Room layout arrangement? It is better to report the results of [21] in Tab. 1.
    +  The evaluation scheme is not clear, since the ideal result is deterministic, how does the method cope with the variability of the DDPM output during the evaluation?

+ Typo: Tab. 2 Caption: quantitative


**Limitations:**

The authors discuss some limitations about the performance and demands on the data. I would like to see more discussions about the quality of the results, which can be more inspiring.

---

> ### Author Rebuttal · Authors · 2023-08-08
>
>
> We thank the reviewer for your positive, insightful and valuable comments and suggestions which are very crucial for improving the quality of our manuscript.
>
> ---
>
> **1. Applications of the room layout arrangement and hardness of the jigsaw puzzle solving.**
>
> >The room layout arrangement is an emerging problem for the real estate industry, with big companies like Ricoh, Zillow, and MagicPlan actively pursuing solutions. For instance, there exist mobile applications that capture individual room layouts, necessitating manual integration to produce the full house floorplans. Our method is a step forward toward automating this process. Please also refer to L11-L13, L22-L24, L106-109, and L201-207 of the paper, mentioning the practical impact of the proposed research.
>  >
> >Moreover, tackling the CJP and VJP demands at least several minutes of human effort. In contrast, our method accomplishes this task within mere seconds. We believe extending our method to similar tasks and datasets could assist in addressing formidable challenges.
> >
> >We also believe that extending our method to other similar tasks and datasets can help solve overwhelming tasks such as reassembling shredded documents or restoring fragmented objects and images, and potentially even more exciting future directions, such as designing electrical boards based on the provided components.
> ---
> **2. Non-learning based methods for Full MagicPlan (RPLAN).**
>
> >As highlighted by Shabani et al. [20], their method needs several minutes to process a small-scale house. However, their processing time explodes exponentially to approximately one hour for a house with seven rooms and even a few days for a house with ten rooms.
> >
> >Analytically, suppose each room has two doors, the number of possible room arrangements by aligning pairs of doors is equal to the number of spanning trees in a complete graph with $n$ vertices, which is $n^{(n-2)}$.  The count becomes approximately 1k, 262k, and 100 million for a floorplan of 6, 8, and 10 rooms, respectively. This demonstrates the severe scalability challenge of non-learning based methods.
>
> **3. The canonical diffusion denoising probabilistic model requires quite some denoising steps.**
>
> >We thank for the great comment. Our current method utilizes the standard diffusion denoising probabilistic models, which are slow and require many sampling steps. However, we rather consider the use of diffusion models to be a distinct advantage of our approach. This is because any advancements in accelerated sampling techniques and efficient diffusion models, which are currently a vibrant area of research, will be directly applicable to our method. This aspect is beyond the scope of our paper due to space constraints and the diverse array of available sampling approaches.
>
> ---
>
> **4. It seems better to derive more analysis on ablation studies.**
>
> >We thank for the interesting questions. We will add the following analysis to the main paper.
> >- Matching loss on doors and corners: One reason can be that the corners of rooms within the floorplans do not necessarily align. For instance, a wall from the dining hall might divide two rooms, thereby not conveying significant high-level information. In contrast, doors hold more informative value as they connect the rooms.
> >- Feature redundancy: While the redundancy in the diffusion process can be considered as adding more randomness and possibly more augmentation, as mentioned by the reviewer, it also serves us in two ways. 1) It helps in the model's architecture design by providing the ability to establish explicit connections among corners within the transformer, for example by utilizing Polygon Self Attention. 2) During the final prediction stage, this redundancy can be beneficial by allowing the averaging or voting of predictions akin to an ensemble model.
>
> ___
>
> **5. Comparison with Lambert et al. [21] for Layout arrangement.**
>
> >While the high-level problem is the same, Lambert et al. [21] assume that the panorama images are given and have some overlaps. They leverage this information to generate Aligned Bird's Eye View Texture Maps, subsequently employing these maps for global pose estimation. However, this is not possible in our targetted application domains, because in a mobile application such as MagicPlan, panorama images cannot be uploaded to the cloud due to the privacy/security concerns as well as to minimize the amount of data transfer from a mobile device (see L296-300). Our approach requires only room layouts that are extremely compact and does not convey any private information.
>
> ---
> **6. How does the method cope with the variability of the DDPM output during the evaluation?**
>
> >We thank for the question and will clarify the following in the paper. We run our system five times and report the mean.
>
> ---
> **7. Typo in Tabel. 2 Caption.**
> >We thank for the typo check and will make a correction.

---

> > ### Comment · Reviewer_zazU · 2023-08-16
> >
> > I thank the authors for clarifying my concerns. I still think the method is reasonable but the motivation is not so sound for me. For example, if users of MaginPlan can take time to draw a room layout, roughly assembling different rooms will not be a problem. The authors did mention some applications that are more interesting and reasonable in their responses. It would be interesting to see some of those applications. For scaling to more complex cases, it would be more convincing to add more complex results in the paper, for example, examples with dozens of pieces. Besides, it is also important to add some discussions about the low-quality results of the present method. I believe this will be interesting to the readers.

---

> > > ### Author Response · Authors · 2023-08-18
> > >
> > >
> > >
> > > We again thank the reviewer for valuable input regarding the experiments, comments, and responses.
> > > We are glad that the reviewer's concerns have been clarified. We further respond to points raised in the discussion.
> > >
> > > ---
> > > **1. For scaling to more complex cases, it would be more convincing to add more complex results in the paper, for example, examples with dozens of pieces.**
> > >
> > > We thank the reviewer for the question. To further address the reviewer’s concern, we created a new version of the Voronoi dataset including 200K samples for training and 2K samples for the test, and we increased the number of pieces to the range of 15 to 50. The obtained metrics for our model are 65.48, 70.40, and 55.31 respectively for Overlap, Precision, and Recall.
> > >
> > > ---
> > >
> > > **2. Besides, it is also important to add some discussions about the low-quality results of the present method. I believe this will be interesting to the readers.**
> > >
> > > We thank the reviewer for the suggestion. While we have provided some failure samples in Fig. 5 of supplementary, and also the first figure of the attached PDF, we will add more cases along with discussion in the final manuscript.
> > >
> > > ---
> > > **3. If users of MaginPlan can take time to draw a room layout, roughly assembling different rooms will not be a problem.**
> > >
> > > Creating an efficient data capture pipeline for the masses significantly differs from data acquisition in controlled research environments. Operators are burdened with multiple intricate steps and lengthy instructions. These steps involve capturing images, annotating layouts and  details like addresses, floor and unit numbers, room types, cardinal headings (NSEW), presence of obstacles, and key architectural elements such as windows, doors, and fixtures like basins, bathtubs, and laundry machine bases.
> > >
> > >  Our collaboration extends across seven global companies focused on applying computer vision techniques in real estate and construction. Automation of as many steps as possible through robust techniques is paramount. Additionally, there's ongoing research to automatically extract room layouts from panoramic images, further enhancing the automation process.
> > >
> > > By combining our research and leveraging these advancements, a fully automated system is on the horizon. This perspective will be emphasized in our paper, reinforcing the significance of our real-world applicable research.
> > >
> > > ---
> > > **4. The authors did mention some applications that are more interesting and reasonable in their responses. It would be interesting to see some of those application.**
> > >
> > > We agree with the reviewer that our paper can motivate several interesting future applications. However, it is worth noting that in references such as Lambert et al. ECCV 2022 [21] (Room arrangement), Harel et al. CVPR 2021 [6] (Crossing cut puzzles), and Shabani et al. ICCV 2019 [20] (Room arrangement), a single task has been addressed. In comparison, our approach not only excels in tackling those tasks more effectively and efficiently, but also introduces an additional layer of complexity through the incorporation of more intricate tasks, such as Voronoi puzzles.

---

> > > > ### Comment · Reviewer_zazU · 2023-08-20
> > > >
> > > > Thank you. The responses address some of my concerns, especially the application of more complex cases.

---

### Official Review · Reviewer_TNRQ · 2023-07-02

**Soundness:** 3 good
**Presentation:** 3 good
**Contribution:** 3 good
**Rating:** 7
**Confidence:** 4

**Summary:**

This paper introduces a novel approach to puzzle solving and room floorplan arrangement by utilizing a conditional generation process based on the denoising diffusion model. The model effectively reconstructs the original polygonal coordinates, representing the spatial arrangement, during the reverse diffusion process. Notably, the proposed method demonstrates robustness to data noises, which enhances its practical applicability.

In order to train the diffusion model, the authors introduce two new datasets: a synthetic jigsaw dataset and a real floorplan dataset with room layout pieces.

The experimental results showcase outstanding performance in both puzzle solving and room floorplan arrangement tasks.

**Strengths:**

1. The utilization of positional information as the signal in the diffusion process is a relatively novel idea.
2. Introduces an interesting approach by framing spatial arrangement, puzzle solving and floorplan registration tasks as a conditional generative process.

**Weaknesses:**

1. The paper exhibits a writing problem where the description of the jigsaw solving and layout arrangement tasks lacks strong connections.
2. It would be beneficial to provide more detailed explanations about the jigsaw part directly in the main paper.
3. The training of the diffusion model requires a large amount of data.

**Questions:**

1. Can the proposed method handle puzzles that are cut into squares?
2. Can a baseline be composed of TransVector-based approaches with multiple iterations of optimization?
3. Could you provide more details on the "Averaging/voting" process in Figure 2 of the supplement?
4. How does the computational cost of the proposed method compare to existing methods?

---

> ### Author Rebuttal · Authors · 2023-08-08
>
>
> We thank the reviewer for your positive, insightful and valuable comments and suggestions which are very crucial for improving the quality of our manuscript.
>
> ---
> **1. Additional details of the jigsaw part in the main paper and lack of connection between layout arrangement and jigsaw puzzles.**
>
> >We thank for the comment and will do our best to clarify the details of the layout arrangement and jigsaw puzzle tasks. Both tasks assemble scattered geometric pieces into a single coherent shape, an image, or a floorplan of a functional house. Careful spatial reasoning is the key to the success of a method. We will clarify this in the paper and would appreciate any further specific feedback/comments on the clarity.
>
> ---
>
> **2. The training of the diffusion model requires a large amount of data.**
>
> >We agree with the reviewer, where one of our key contributions is the introduction of a new large-scale real-world dataset, MagicPlan, which helps us train our diffusion-based model. This contribution is precious given the scarcity of substantial real estate data, owing to privacy and licensing constraints. In addition, this challenge does not apply to Jigsaw puzzle tasks whose datasets are often synthetic and scale up easily.
>
> ---
>
> **3. Can the proposed method handle puzzles that are cut into squares?**
>
> >We thank for the question and have conducted experiments to show that our method can solve square jigsaw puzzles.
> >
> >Specifically, we have solved 3x3 pictorial square puzzles with a similar implementation as for the pictorial CJP. Our model achieves the following numbers:
> >- 90.98% in *Direct Comparison*, which denotes the portion of puzzle pieces within the reassembled puzzle that have been positioned correctly,
> >- 88.87% in *Neighbour* score, meaning the percentage of paired neighboring pieces that are correct,
> >- 74.54% in *Perfect Reconstruction*, which measures the percentage of perfectly reassembled puzzles.
> >
> >We want to emphasize that our approach randomly initializes and solves the arrangements in a continuous space. In contrast, prior methods such as [35,36] focus solely on a subset of potential discrete permutations. For instance, Table 4 in reference [35] displays a maximum of 1000 permutations selected from the entire pool of 9! (~363k) possible permutations. Even with this reduced set, the challenge posed by the number of permutations persists, especially compared to the results from consideration of a mere 10 or 100 permutations. A few qualitative samples are provided in the attached pdf file. We will add this to the main paper.
>
> ___
>
> **4. Can a baseline be composed of TransVector-based approaches with multiple iterations of optimization?**
>
> >We thank for the suggestion and have implemented this new baseline as an additional experiment. Drawing inspiration from the iterative approach employed by Housegan++ [56] during training, we integrated a 50% probability for each room to remain fixed. In cases where rooms were designated as fixed, we input the ground truth position (GT position) to the network, enabling the network to learn the task of arranging non-fixed rooms by leveraging information from the fixed ones.
> >
> >During the testing phase, previously predicted locations were introduced to the network with a 50% probability per room, effectively offering potential input constraints and facilitating iterative design refinement. This process was repeated a total of 10 times. Consequently, the performance improved, yielding MPE of 39.12 and GED of 2.18 on the full RPLAN dataset, while the full MagicPlan dataset reflected 47.85 and 5.49, respectively for MPE and GED. Despite noteworthy improvements, the results remained significantly inferior to our diffusion-based approach. We will add this to the paper.
>
> ---
>
> **5. Could you provide more details on the "Averaging/voting" process in Figure 2 of the supplement?**
>
> >We thank for the question and will clarify the following details in the paper. For puzzle solving, the determination of the location and rotation of each piece involves computing the average location and rotation of its corners. However, in the context of room layout arrangement, where rotations adhere exclusively to the Manhattan directions (0, 90, 180, and 270 degrees), a voting system for rotation is employed. This includes assigning each corner a vote, aligned with the rotation closest to its predicted value. The final rotation of the piece corresponds to the angle that accumulates the highest number of votes. For position, we use a similar approach to puzzle solving. Predicted positions for all corners within a room are averaged to establish the position of the room within the floorplan.
> >
> >When visualizing the diffusion process, there are two options: utilizing the averaged output of the entire piece for corner positions, or solely relying on the output of each corner itself. Figure 2 in the supplementary material and the video from 00:57 to 1:26 provides both visualizations.
>
> ---
> **6. How does the computational cost of the proposed method compare to existing methods?**
>
> >The diffusion model and TransVector implementations each utilize approximately 4 million parameters. On the other hand, Harrel operates as a non-learning-based method, devoid of any parameters to be learned. In the case of Shabani et al. [20], their approach encompasses a blend of a heuristic technique for generating potential candidates alongside a deep learning model for scoring said candidates. While the model itself incorporates a lesser number of parameters, the heuristic component demands notably more time due to the generation of all possible candidates via door connections. The training and the inference time of our method and the competing ones are given at L183-187 and L234-238 of the paper.

---

> > ### Comment · Reviewer_TNRQ · 2023-08-12
> >
> > I appreciate your intention to clarify the details of both the jigsaw puzzle and layout arrangement tasks. Allocating some space to underscore the shared concept between these tasks —"assemble scattered geometric pieces into a single coherent shape, an image, or a floorplan"- would enhance the overall coherence of the narrative.
> >
> > Thank you for providing more experimental results that demonstrate the ability of your method to handle square jigsaw puzzles and the new baseline test-time iterated TransVector approach.
> >
> > All of my concerns are addressed.

---

> > > ### Author Response · Authors · 2023-08-13
> > >
> > > We again thank the reviewer for valuable input regarding the experiments, comments, and responses. We'll enhance the explanation of the shared concept in all three tasks and include additional experiments. Please let us know if you have further questions or comments.

---

### Official Review · Reviewer_AMsE · 2023-07-07

**Soundness:** 3 good
**Presentation:** 3 good
**Contribution:** 3 good
**Rating:** 6
**Confidence:** 2

**Summary:**

This paper investigates the use of diffusion models to solve jigsaw puzzles. Puzzle pieces are model as a sequence of corners and the diffusion process consists in adding noise/denoising the position and orientation of the corners, conditionally to the original piece shapes. To have fragments snap into place, an additional loss for corner matching is added. The diffusion process is test on synthetic puzzles as well as on a floorplan dataset and shows that the proposed method in improving over the existing literature.

**Strengths:**

Given the combinatorial nature of jigsaw puzzles, using a denoising diffusion process, which runs in a fixed number of steps, is a promising idea. It opens the door of using diffusion process to solve discrete combinatorial tasks, which is an area where learning based method are still struggling to dominate (setting aside Deep MCTS). The execution in this paper is well done, the additional loss for corner matching is sound. The experiments are OK. Code was provided.

**Weaknesses:**

Not much!
- One complaint I have is that the evaluation metrics are not very interpretable. The MPE depends on the size of the puzzle and the pieces (a 2 pix error is not the same on a 5 pix fragment and on a 150 pix fragment), GED is even worse and precision/recall tend to saturate on the proposed datasets. Also, pictorial CJP have no quantitative results mentioned.
- I was having the impression that the literature on jigsaw puzzle solving using deep learning is a bit sparse ([26, 35, 36]), given the number of relevant papers that a google scholar search returns when querying "deep learning jigsaw puzzle". More comparison with how deep learning was used to solve the problem would have be nice to highlight the radical difference this approach proposes.

**Questions:**

- Is it possible to solve the regular n x n jigsaw puzzle (for example by assigning to each fragment its closest fixed position) and compare to regular method that solve the combinatorial problem (like [35,36] and similar methods) using fragment position accuracy and puzzle solving accuracy? This would allow for much more interpretable results.
- What are the score for pictorial CJP?

**Limitations:**

The paper addresses limitations in the conclusion by mentioning the requirement of large scale training sets.

---

> ### Author Rebuttal · Authors · 2023-08-08
>
> We thank the reviewer for your positive, insightful, and valuable comments.
>
> ---
> **1. Interpretability of the metrics.**
>
> >We thank the reviewer for pointing this out. While these metrics might not be deemed ideal, it worth to mention that for layout arrangement, as every pixel is equivalent to a real world distance (measured in meters/feet), which can be determined using the room layout scale of the house, MPE can easily converted to real-world distance, as highlighted by Shabani et al. [20]. In addition, to further address the reviewer's concern, we have devised a new **"Weighted MPE" (WMPE)** metric, which normalizes the error per room/piece by the square root of the room/piece area:
> >$
> \text{WMPE} =\frac{1}{N} \sum_{i=0}^{N} \frac{\text{Positional Error of } x_{i}}{\sqrt{\text{ Area of }x_{i} }}
> $
> >
> >The WMPE metric scores are 0.109 (ours) and 0.451 (TransVector) for the full RPLAN dataset, and 0.341 (ours) and 0.564 (TransVector) for the full MagicPlan dataset. We will add the results to the paper.
>
> >The GED metric was borrowed from Nauata et al. [56] to assess room connectivity, which is established through doors/frames, stands as a pivotal factor in house design. The precision/recall metric has been the standard in the recent jigsaw puzzle literature [6,26].
>
> ---
> **2. Quantitative results of CJP.**
>
> >Due to the time complexity of Harel et al. especially in the case of pictorial puzzles, they did not report any number for their method on pictorial data. For the same reason, in order to compare Harel et al. to ours, we calculated the metrics on 20 samples for pictorial CJP. Our method achieves 0.953, 0.978, and 0.930, respectively for, IoU, Precision, and Recall, compared to 0.935, 0.934, and 0.947 of Harel et al. Thank you for pointing out, we will add the details to the paper.
>
>
> ---
> **3. Additional related works on jigsaw puzzles.**
>
> >We thank for the suggestion. We will add and discuss more references on deep-learning based jigsaw puzzle papers. However, the majority of these papers tackle square puzzles and cannot be compared on our tasks [25]. Our work pursues a broader and more adaptable approach for more challenging tasks, and needs to compare against a few similarly adaptable approaches (e.g., [Harel et al.] or [Shabani et al.]).
>
> ---
> **4. Is it possible to solve the regular n x n jigsaw puzzle?**
>
> >Yes, although the focus of our paper is mainly on the geometry side of the puzzles, our method can be used for pictorial square puzzles as well. In this regard, we did an experiment on solving 3x3 pictorial square puzzles with similar implementation details as the pictorial CJP. Our model achieves the following metrics:
> >- 90.98% in *Direct Comparison*, which denotes the portion of puzzle pieces within the reassembled puzzle that have been positioned correctly,
> >- 88.87% in *Neighbour* score, meaning the percentage of paired neighboring pieces that are correct,
> >- 74.54% in *Perfect Reconstruction*, which measures the percentage of perfectly reassembled puzzles.
> >
> >We need to emphasize that our approach tackles puzzles with randomly initialized pieces in a continuous space. In contrast, prior methods such as [35,36] focus solely on a subset of potential discrete permutations. For instance, Table 4 in reference [35] displays a maximum of 1000 permutations selected from the entire pool of 9! (~363k) possible permutations. Even with this reduced set, the challenge posed by the number of permutations persists, especially compared to the results from consideration of a mere 10 or 100 permutations. Few qualitative samples are provided in the attached PDF file. We will add this to the main paper.

---

> > ### Comment · Reviewer_AMsE · 2023-08-15
> >
> > The rebuttal addresses my concerns.

---

> > > ### Author Response · Authors · 2023-08-18
> > >
> > > We again thank the reviewer for valuable input regarding the experiments, comments, and responses.  We are glad that our responses addressed all the reviewer's concerns. We will add discussed clarifications and experiments to the final manuscript.

---

### Official Review · Reviewer_qFVS · 2023-07-19

**Soundness:** 3 good
**Presentation:** 2 fair
**Contribution:** 2 fair
**Rating:** 5
**Confidence:** 4

**Summary:**

The paper presents a diffusion based method to tackle jigsaw puzzle solving task. This task has applications in artwork restoration, room layout estimation, etc. The paper also introduces a room layout and arrangement dataset. The authors compare their work with previous state of the art approaches and a transformer based baseline, and outperforms all these methods.

**Strengths:**

1. The paper presents an interesting approach to tackle jigsaw puzzle solving problem using diffusion models. The corner coordinates and rotation of puzzle pieces are used as input, and diffusion model is tasked with predicting the noise in those inputs.
2. Unlike prior methods, this method is not limited to simple puzzles or require pairwise comparison between puzzle pieces, which makes it more efficient.
3. The method achieves better performance compared to previous state of the art methods and a transformer based baseline that directly predicts the output instead of doing iterative denoising.
4. Ablations show that all new modules/loses introduced in the paper contribute towards the final performance.

**Weaknesses:**

### Paper clarity comments

1. Section 3 is a bit hard to follow.
2. What does line 140 mean: “position/rotation of the rth room/piece stored at the ith corner”? Does it mean the piece that has corner i as one of the corners?

### Missing ablation
3. Authors state that redundant representation helps their system, and have shown one ablation for it. However, it would be interesting to see an ablation that shows how the method performs when the the position and orientation estimation of the piece are not concatenated with all the corners but passed as separate input.

### Discrepancy between paper and supplementary
4. Some discrepancy between figure and supplementary video: In fig 2, it seems like that random noise is added to each corner independently and that noise changes the change of each piece. In the supplementary video, however, it seems that same random noise is added to all the corners of each piece, which results in shape of each piece staying consistent. So which one is it?
5. If random noise is added to each corner independently for each piece, how do authors maintain the ordering of the corners when passing it to transformers? Won’t de-noising process change the ordering of corners?

**Questions:**

### Copying my comments from weakness section

1.  Authors state that redundant representation helps their system, and have shown one ablation for it. However, it would be interesting to see an ablation that shows how the method performs when the the position and orientation estimation of the piece are not concatenated with all the corners but passed as separate input.

2. Some discrepancy between figure and supplementary video: In fig 2, it seems like that random noise is added to each corner independently and that noise changes the change of each piece. In the supplementary video, however, it seems that same random noise is added to all the corners of each piece, which results in shape of each piece staying consistent. So which one is it?

3. If random noise is added to each corner independently for each piece, how do authors maintain the ordering of the corners when passing it to transformers? Won’t de-noising process change the ordering of corners?


**Limitations:**

Yes, authors have addressed the limitations well

---

> ### Author Rebuttal · Authors · 2023-08-08
>
> We thank the reviewer for your positive, insightful and valuable comments and suggestions which are very crucial for improving the quality of our manuscript.
>
> ---
>
> **1. Section 3 is a bit hard to follow**
>
> >We thank for the comment and will do our best to clarify our writing. We would appreciate any further specific comments or suggestions.
> ---
>
> **2. What does line 140 mean: ''position/rotation of the rth room/piece stored at the ith corner''? Does it mean the piece that has corner i as one of the corners?**
>
> > We thank for the question and will clarify in the paper. ($r$) denotes the index of a room/piece in a puzzle. For example. ($r$) is either 1, 2, or 3 for a puzzle of 3 pieces. Similarly, ($i$) denotes the index of a corner in a room/piece. Therefore ($C^r_i$) denotes the i-th corner of the r-th room/piece. Note that our approach infers the position/rotation information at every corner of a room/piece and uses their average to determine the final position/rotation of the room/piece.
>
> ---
>
> **3. it would be interesting to see an ablation that shows how the method performs when the the position and orientation estimation of the piece are not concatenated with all the corners but passed as separate input.**
>
> >That is indeed an interesting experiment. In order to study network performance while considering position and orientation as separate inputs, we conducted two experiments. In the first experiment, we utilized a single input token per piece to predict both piece position and rotation of that piece, while incorporating corners information as separate tokens. In the second set of experiments, we employed one input token for each corner to predict the position and rotation of each piece, with corner information again represented as distinct tokens. We applied both of these methods to the MagicPlan dataset.
> >
> >For the first setup, we achieved MPE and GED values of 49.91 and 4.96, respectively. Similarly, for the second setup, the corresponding values were 42.45 for MPE and 3.22 for GED. The performance drop is significant in the first setup, and although the second setup yields more comparable results, it still demonstrates lower performance when compared to our setup (MPE/GED of 40.81/3.09). Additionally, it involves twice the number of input tokens compared to our setup, resulting in higher computational costs. We will provide details of this experiment in the paper.
>
>
> ---
>
>
>
> **4. It seems that same random noise is added to all the corners of each piece.**
>
> >Different noises (but from the same Gaussian distribution) are added to different corners, with each corner contributing to a prediction for the position or rotation of the corresponding piece. The final prediction is derived by averaging the predictions from all corners of a piece. Nevertheless, when visualizing the diffusion process, there are two options: utilizing the averaged output of the entire piece for corner positions, or solely relying on the output of each corner itself. Figure 2 in the supplementary material and the video from 00:57 to 1:26 effectively illustrate this concept.
>
> ___
>
> **5. How do authors maintain the ordering of the corners when passing it to transformers?**
>
> >We thank for the question. The feature embedding contains the corner index information (the second term of Eq. 3 in Section 4.2), which allows our system to keep track of the order of corners. We will clarify our explanation at Lines 156-158 of the main paper.

---

> > ### Comment · Reviewer_qFVS · 2023-08-16
> >
> > I thank the authors for their response and for addressing my concerns. I have increased my final score to 5.

---

> > > ### Author Response · Authors · 2023-08-18
> > >
> > > We again thank the reviewer for valuable input regarding the experiments, comments, and responses.
> > > We sincerely thank the reviewer for updating the score and We are glad that our responses addressed all the reviewer's concerns. We will make sure to incorporate your suggestions by adding further clarification and experiments in our final manuscript.

---

### Official Review · Reviewer_c71V · 2023-07-19

**Soundness:** 3 good
**Presentation:** 4 excellent
**Contribution:** 3 good
**Rating:** 6
**Confidence:** 2

**Summary:**

This paper introduces a diffusion model for solving 3 different spatial puzzle tasks: cross-cut jigsaw, voronoi jigsaw, and room layout arrangement. Their method achieves SOTA results while being much faster than previous works, allowing them to handle larger puzzles than previous methods could. They also demonstrate greater robustness to noisy data inputs compared to previous methods, despite not being trained on noisy data. Ablations show the contributions of proposed losses and network components. They also present two new datasets: a synthetic one for the Voronoi puzzle task, and a real room layout one from MagicPlan.

**Strengths:**

- The paper is overall well-written and easy to understand.
- In terms of originality, this paper is the first to tackle these kinds of layout arrangement problems using diffusion models.
- The proposed method is sound and clearly described.
- The method is a significant improvement over existing methods while using a very different approach.

**Weaknesses:**

- No major weaknesses
- At zero noise level (Figs 5 and 6), the proposed method seems to be less precise at aligning the pieces compared to Harel et al. (while the general layout is correct, there are small alignment errors). Of course, Harel et al. fails for more difficult cases, but for these easier cases, it seems slightly more accurate than the proposed method. (I am however not very familiar with this task so I am not sure how important these small inaccuracies are in practice.)
- Some details are unclear (see “Questions”)

**Questions:**

- As shown in Fig. 7 of the supplementary PDF, the method produces slightly different results due to stochasticity; how many samples were generated when computing the results in Tables 1 and 2?
- L243 says that TransVector achieves a better MPE score than Shabani et al, but that doesn’t appear to be the case in Table 1?
- In Table 2, are the CJP and VJP results obtained using the same model, or is a separate model trained for each task?
- For Table 1, separate models are trained for MagicPlan and RPLAN, but does a model trained on MagicPlan exhibit some generalization to RPLAN and vice versa?

**Limitations:**

Limitations were discussed adequately.

---

> ### Author Rebuttal · Authors · 2023-08-08
>
> We thank the reviewer for your positive, insightful and valuable comments and suggestions which are very crucial for improving the quality of our manuscript.
>
>
> ---
> **1. At zero noise level (Figs 5 and 6), the proposed method seems to be less precise at aligning the pieces compared to [6].**
> >We agree that Harel et al. achieves slightly more precise alignment through their heuristic method in the case of zero noise.
> However, this is not an issue. We conducted an experiment (see the attached pdf Fig. 2) showing that our approach easily achieves equally precise alignment by a simple post-processing heuristic, which is a minor variant of a loop merging process of an existing work [1]. Our method is also noticeably faster than [6]. We will add the details of the post-processing step and more experimental results to the paper.
>
> >[1] Chen, Jiacheng, et al. "Floor-sp: Inverse cad for floorplans by sequential room-wise shortest path." Proceedings of the IEEE/CVF International Conference on Computer Vision. 2019.)
>
> ---
>
> **2. How many samples were generated when computing the results in Tables 1 and 2?**
>
> >We thank for the question and will clarify in the paper. We ran our system 5 times and reported the mean.
>
> ---
>
> **3. Typo on line 243.**
>
> >We apologize for the oversight and thank you for bringing that to our attention. That sentence is indeed a typo, and we will drop the phrase from the text.
> ___
>
> **4. Are the CJP and VJP results obtained using the same model, or is a separate model trained for each task?**
>
> >We thank for the question and will clarify in the paper. The models are independent and trained separately.
>
> ---
> **5. Does a model trained on MagicPlan exhibit some generalization to RPLAN and vice versa?**
>
> >We thank for a great question. We conducted additional experiments, which will be added to the paper. Concretely, we trained our model with MagicPlan and tested it with RPLAN. (MPE/GED) scores are (15.62/1.90), compared to (10.55/0.97) when trained and tested with RPLAN. Similarly, (MPE/GED) scores are (48.48/4.68) when trained with RPLAN and tested with MagicPlan, compared to (40.81 and 3.09) when trained and tested with MagicPlan. The performance drops are understandable, given that Magicplan encompasses real-world data contributed by users, while RPLAN contains synthetic noise-free floorplans, designed by professional architects. It is worth noting that our cross-dataset results are still noticeably better than in-dataset results by the TransVector baseline.

---

> > ### Comment · Reviewer_c71V · 2023-08-16
> >
> > I thank the authors for the clarifications and additional experiments. The rebuttal addressed all of my concerns.

---

> > > ### Author Response · Authors · 2023-08-18
> > >
> > > We again thank the reviewer for valuable input regarding the experiments, comments, and responses. We are glad that our responses addressed all the reviewer's concerns. We will incorporate the discussed clarifications and experiments into the final manuscript.

---

### Author Rebuttal · Authors · 2023-08-08

We thank all the reviewers for their valuable and insightful comments. We are also grateful to the reviewers for their positive comments on our work. We have addressed the reviewers points in our individual responses to each reviewer, and please let us know if there are any new questions.

---

### Decision · Program_Chairs · 2023-09-21

**Decision:**

Accept (spotlight)

**Comment:**

The paper received unanimous positive feedbacks. The paper is well-written and the problem is novel. Some issues related to clarity and applications are resolved after the rebuttal.